# Spin current generation and relaxation in a quenched spin-orbit-coupled Bose-Einstein condensate

Chuan-Hsun Li[1], Chunlei Qu [2,3,4], Robert J. Niffenegger [5,8], Su-Ju Wang [5,9], Mingyuan He [6], David B. Blasing[5], Abraham J. Olson[5], Chris H. Greene[5,7], Yuli Lyanda-Geller[5,7], Qi Zhou[5,7], Chuanwei Zhang[2] & Yong P. Chen[1,5,7]

Understanding the effects of spin-orbit coupling (SOC) and many-body interactions on spin transport is important in condensed matter physics and spintronics. This topic has been intensively studied for spin carriers such as electrons but barely explored for charge-neutral bosonic quasiparticles (including their condensates), which hold promises for coherent spin transport over macroscopic distances. Here, we explore the effects of synthetic SOC (induced by optical Raman coupling) and atomic interactions on the spin transport in an atomic Bose-Einstein condensate (BEC), where the spin-dipole mode (SDM, actuated by quenching the Raman coupling) of two interacting spin components constitutes an alternating spin current. We experimentally observe that SOC significantly enhances the SDM damping while reducing the thermalization (the reduction of the condensate fraction). We also observe generation of BEC collective excitations such as shape oscillations. Our theory reveals that the SOC-modified interference, immiscibility, and interaction between the spin components can play crucial roles in spin transport.

[1] School of Electrical and Computer Engineering, Purdue University, West Lafayette, IN 47907, USA. [2] Department of Physics, The University of Texas at Dallas, Richardson, TX 75080, USA. [3] INO-CNR BEC Center and Dipartimento di Fisica, Università di Trento, Povo 38123, Italy. [4] JILA and Department of Physics, University of Colorado, Boulder, CO 80309, USA. [5] Department of Physics and Astronomy, Purdue University, West Lafayette, IN 47907, USA. [6] Department of Physics, Hong Kong University of Science and Technology, Clear Water Bay, Hong Kong, China. [7] Purdue Quantum Center, Purdue University, West Lafayette, IN 47907, USA. [8] Present address: Lincoln Laboratory, Massachusetts Institute of Technology, 244 Wood Street, Lexington, MA 02421, USA. [9] Present address: J. R. Macdonald Laboratory, Department of Physics, Kansas State University, Manhattan, KS 66506, USA. Correspondence and requests for materials should be addressed to Y.P.C. (email: yongchen@purdue.edu)

Spin, an internal quantum degree of freedom of particles, is central to many condensed matter phenomena such as topological insulators and superconductors[1,2] and technological applications such as spintronics[3] and spin-based quantum computation[4]. Recently, neutral bosonic quasiparticles (such as exciton-polaritons and magnons) or their condensates[5–7] have attracted great interest for coherent manipulation of the spin information. For example, spin currents have been generated using exciton-polarions[8] and excitons[9] in semiconductors and magnons[10,11] in a magnetic insulator. In spin-based devices, SOC and many-body interactions are key factors for spin current manipulations. SOC can play a particularly crucial role as it may provide a mechanism (such as spin Hall effect) to control the spin, however, it can also cause spin (current) relaxation, leading to loss of spin information. Studying the effects of SOC and many-body interactions on spin relaxation is thus of great importance but also challenging due to uncontrolled disorders and the lack of experimental flexibility in solid state systems.

Cold atomic gases provide a clean and highly-controllable[12] platform for simulating and exploring many condensed matter phenomena[12–16]. For example, the generation of synthetic electric[17] and magnetic[18] fields allows neutral atoms to behave like charged particles. The synthetic magnetic and spin-dependent magnetic fields have been realized to demonstrate respectively the superfluid Hall[19] and spin Hall effects[20] in BECs. The creation of synthetic SOC in bosonic[21–25] and fermionic[26–29] atoms further paves the way to explore diverse phenomena such as topological states[30] and exotic condensates and superfluids[16,31–35]. Here, we study the effects of one-dimensional (1D) synthetic SOC on the spin relaxation in a disorder-free atomic BEC using a condensate collider, in which the SDM[36] of two BECs of different (pseudo) spin states constitute an alternating (AC) spin current. The SDM is initiated by applying a spin-dependent synthetic electric field to the BEC via quenching the Raman coupling that generates the spin-orbit-coupled (SO-coupled) band structure. Similar quantum gas collider systems (without SOC[37–41]) have been used to study physics that are difficult to access in other systems.

Charge or mass currents are typically unaffected by interactions between particles because the currents are associated with the total momentum that is unaffected by interactions. In contrast, spin currents can be intrinsically damped due to the friction resulting from the interactions between different spin components. In electronic systems, such a friction has been referred to as the spin Coulomb drag[42,43]. In atomic systems, previous studies have shown that a similar spin drag[44,45] also exists. Even in the absence of SOC, the relaxation of spin currents can be nontrivial due to, for example, interactions[36,39,46–49] and quantum statistical effects[45,50]. In one previous experiment[20], bosonic spin currents have been generated in a SO-coupled BEC using the spin Hall effect. However, how the spin currents may relax in the presence of SOC and interactions has not been explored. Here, we observe that SOC can significantly enhance the relaxation of a coherent spin current in a BEC while reducing the thermalization during our experiment. Moreover, our theory, consistent with the observations, discloses that the interference, immiscibility, and interaction between the two colliding spin components can be notably modified by SOC and play an important role in spin transport.

## Results

**Experimental setup.** In our experiments, we create 3D $^{87}$Rb BECs in the $F = 1$ hyperfine state in an optical dipole trap with condensate fraction $f_c > 0.6$ containing condensate atom number $N_c \sim 1$–$2 \times 10^4$. As shown in Fig. 1a, counter-propagating Raman lasers with an angular frequency difference $\Delta\omega_R$ couple bare spin and momentum states $|\downarrow, \hbar(q_y + k_r)\rangle$ and $|\uparrow, \hbar(q_y - k_r)\rangle$ to create synthetic 1D SOC (so called equal Rashba–Dresselhaus SOC) along $\hat{y}$[24], where the bare spin states $|\downarrow\rangle = |m_F = -1\rangle$ and $|\uparrow\rangle = |m_F = 0\rangle$ are Zeeman split by $\hbar\omega_Z \approx \hbar\Delta\omega_R$ using a bias magnetic field $\mathbf{B} = B\hat{z}$. Here, $\hbar k_\downarrow = \hbar(q_y + k_r)$ ($\hbar k_\uparrow = \hbar(q_y - k_r)$) is the mechanical momentum in the $y$ direction of the bare spin component $|\downarrow\rangle$ ($|\uparrow\rangle$), where $\hbar q_y$ is the quasimomentum. The photon recoil momentum $\hbar k_r = 2\pi\hbar/\lambda$ and recoil energy $E_r = \hbar^2 k_r^2/(2m)$ are set by the Raman laser at the "magic" wavelength $\lambda \sim 790$ nm[51], where $\hbar$ is the reduced Planck constant and $m$ is the atomic mass of $^{87}$Rb. The $|m_F = +1\rangle$ state can be neglected in a first-order approximation due to the quadratic Zeeman shift (see Methods). The single-particle SOC Hamiltonian, $H_{SOC}$, can be written in the basis of bare spin and momentum states

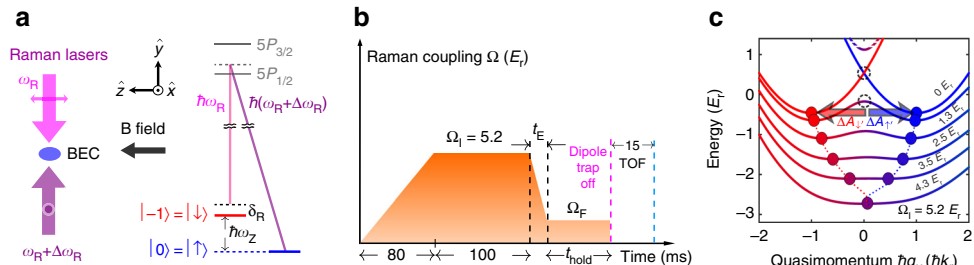

**Fig. 1** Experimental setup and timing diagram used for the spin-dipole mode (SDM) experiments. **a** Linearly polarized Raman beams with orthogonal polarizations (indicated by the double-headed arrows along $\hat{z}$ and $\hat{x}$) counter-propagating along $\hat{y}$ couple $m_F$ hyperfine sublevels (bare spin states) of $^{87}$Rb atoms. The sublevels are Zeeman split by $\hbar\omega_Z \approx \hbar\Delta\omega_R = h \times (3.5$ MHz) using a bias magnetic field $\mathbf{B} = B\hat{z}$, which controls the Raman detuning $\delta_R = \hbar(\Delta\omega_R - \omega_Z)$. **b** Experimental timing diagram: Raman coupling $\Omega$ (with an experimental uncertainty of <10%) is slowly ramped up in 80 ms to an initial value $\Omega_I$ and held for 100 ms to prepare the BEC around the single minimum of the ground band at $\Omega_I$ as shown in **c**. Then, $\Omega$ is quickly lowered to a final coupling $\Omega_F$ in time $t_E$ and held for some time $t_{hold}$, during which we study the dynamics of the BEC in the dipole trap. Subsequently, the atoms are released for absorption imaging after a 15 ms time of flight (TOF), at the beginning of which a Stern–Gerlach process is performed for 9 ms to separate atoms of different bare spin states. **c** The ground band (solid lines) of synthetic SOC is calculated for a few representative $\Omega$ at $\delta_R = 0$. A higher band calculated for $\Omega = 1.3$ $E_r$ is shown as dashed lines. The colors indicate the spin compositions, with red for $|\downarrow\rangle$ and blue for $|\uparrow\rangle$. The ground band minima in quasimomentum marked by dots are identified with spin-dependent vector potentials ($A_\sigma$), which shift in opposite directions as $\Omega$ is lowered into the double minima regime during $t_E$. This generates spin-dependent synthetic electric fields $E_\sigma$ and thus excites the SDM and an AC spin current along the SOC direction in a trapped BEC. The upper (lower) dashed circle represents the region around $q_y = 0$ in the double minima band at an exemplary $\Omega_F = 0$ ($\Omega_F = 1.3$ $E_r$), from which the two (dressed) spin components of the BEC roll down towards the corresponding band minima in response to the application of $E_\sigma$

$\left\{\left|\downarrow, \hbar\left(q_y + k_r\right)\right\rangle, \left|\uparrow, \hbar\left(q_y - k_r\right)\right\rangle\right\}$ as[21]:

$$H_{SOC} = \begin{pmatrix} \frac{\hbar^2}{2m}\left(q_y + k_r\right)^2 - \delta_R & \frac{\Omega}{2} \\ \frac{\Omega}{2} & \frac{\hbar^2}{2m}\left(q_y - k_r\right)^2 \end{pmatrix} \quad (1)$$

where $\Omega$ is the Raman coupling (tunable by the Raman laser intensity), $\delta_R = \hbar(\Delta\omega_R - \omega_Z)$ is the Raman detuning (tunable by $B$) and is zero in our main measurements (see Methods). A dressed state is an eigenstate of Eq. (1), labeled by $q_y$, and is a superposition of bare spin and momentum states. The $q_y$-dependent eigenvalues of (1) define the ground and excited energy bands. When $\Omega$ is below a critical $\Omega_c$, the ground band exhibits double wells, which we associate with the dressed spin up $|\uparrow\,'\rangle$ and down $|\downarrow\,'\rangle$ states. The double minima at quasimomentum $\hbar q_{\sigma\,min}$ can be identified with the light-induced spin-dependent vector potentials $\mathbf{A}_\sigma = A_\sigma \hat{y}$ (controllable by $\Omega$), where $\sigma$ labels $|\uparrow\,'\rangle$ or $|\downarrow\,'\rangle$[20] (see Methods). The double minima merge into a single minimum as $\Omega$ increases beyond $\Omega_c$, as shown in the dashed line trajectories in Fig. 1c.

We prepare a BEC around the single minimum of the ground dressed band at $\Omega_I$ ($= 5.2\ E_r$ for this work) and $\delta_R = 0$ by ramping on $\Omega$ slowly in 80 ms and holding it for 100 ms (Fig. 1b, c, see Methods for details). Then, we quickly lower $\Omega$ from $\Omega_I$ to a final value $\Omega_F$ into the "double minima" regime in time $t_E$. The $t_E = 1$ ms used in this work is slow enough to avoid higher band excitations but is fast compared to the trap frequencies. The dotted lines in Fig. 1c trace the opposite trajectories of $A_{\uparrow'}$ and $A_{\downarrow'}$ during $t_E$. This quench process drives the system across the single minimum to double minima phase transition and generates spin-dependent synthetic electric fields $\mathbf{E}_\sigma = E_\sigma \hat{y} = -(\partial A_\sigma/\partial t)\hat{y} \approx -(\Delta A_\sigma/t_E)\hat{y}$. Consequently, atoms in different dressed spin components move off in opposite directions from the trap center (or from the region around $q_y = 0$ in the quasimomentum space as shown in Fig. 1c as dashed circles for two representative $\Omega_F = 0, 1.3\ E_r$) and then undergo out-of-phase oscillations, thus exciting the SDM and an AC spin current. Approximately equal populations in the two dressed (or bare) spin components are maintained by keeping

$\delta_R = 0$ as $\Omega$ is changed from $\Omega_I$ to $\Omega_F$ (see Methods). After the application of $E_\sigma$, the Raman coupling is maintained at $\Omega_F$ during the hold time ($t_{hold}$). We then abruptly turn off both the Raman lasers and the dipole trap for time of flight (TOF) absorption imaging, measuring the bare spin and momentum composition of the atoms (Fig. 1b). Experiments are performed at various $t_{hold}$ to map out the time evolution in the trap.

**Measurements of the spin-dipole mode (SDM) and its damping.** Figure 2 presents SDM measurements for a bare BEC (at $\Omega_F = 0$) and a dressed (or SO-coupled) BEC (at $\Omega_F = 1.3\ E_r$), with select TOF images taken after representative $t_{hold}$ in the trap. Two TOF images labeled by $t_{hold} = -1$ ms are taken right before the application of $E_\sigma$. In the bare case (Fig. 2a), the images taken at increasing $t_{hold}$ show several cycles of relative oscillations (SDM) between the two spin components in the momentum space, accompanied by a notable reduction in the BEC fraction. We refer to the reduction of condensate fraction in this paper as thermalization. In the dressed case at $\Omega_F = 1.3\ E_r$ (Fig. 2b), despite the fact that $A_\sigma$ are nearly the same as that for the bare case, the SDM is now strongly damped without completing one period. Besides, we observe higher BEC fraction remaining at the end of the measurement compared with the bare case. This can be seen in the narrower momentum distribution of thermal atoms with a more prominent condensate peak in Fig. 2b. From the TOF images, we fit the atomic cloud of each bare spin component (or dominant bare spin component of a dressed spin component) to a 2D bimodal distribution to extract the center-of-mass (CoM) momentum $\hbar k_{\uparrow(\downarrow)}$ or other (dressed) spin-dependent quantities (see Methods). The relative mechanical momentum between the two spin components in the SDM is then determined by $\hbar k_{spin} = \hbar(k_\uparrow - k_\downarrow)$.

Figure 3a–e presents measurements of $\hbar k_{spin}$ versus $t_{hold}$ at various $\Omega_F$. We see that the initial amplitude ($2\hbar k_r$) of $\hbar k_{spin}$ is larger than the width of the atomic momentum distribution ($<\hbar k_r$), and $\hbar k_{spin}$ damps to around zero at later times. The observed $\hbar k_{spin}$ as a function of $t_{hold}$ is fitted to a damped sinusoid $A_0 e^{-t_{hold}/\tau_{damp}} \cos(\omega t_{hold} + \theta_0) + B_0$ (see Methods) to extract the decay time constant $\tau_{damp}$. The SDM damping is then quantified

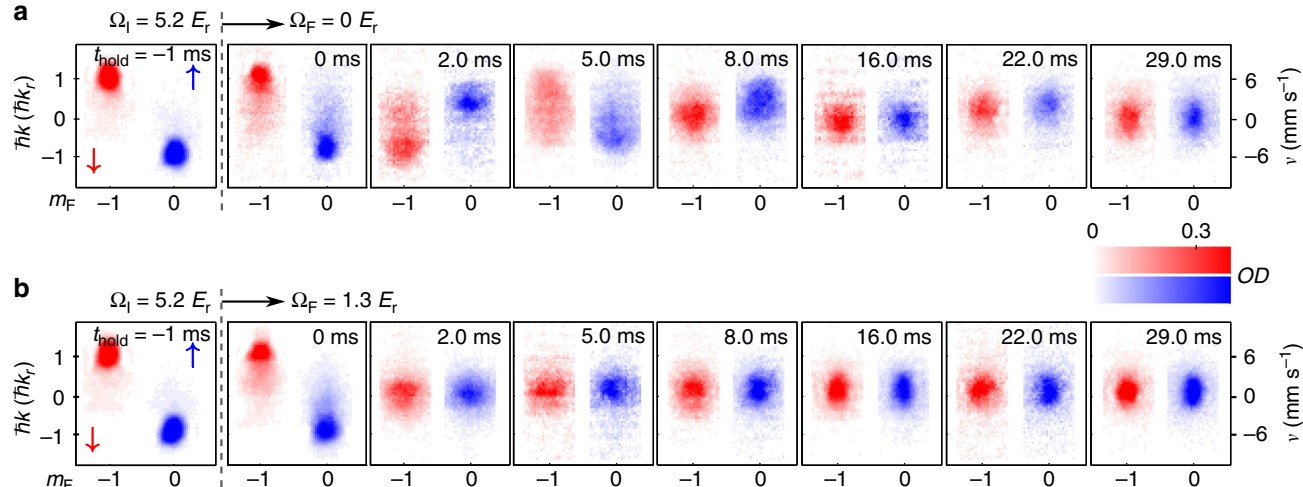

**Fig. 2** SDM of a bare or a dressed BEC. Select TOF images showing the bare spin and momentum compositions of atoms taken after applying spin-dependent synthetic electric fields $E_\sigma$ with $\Omega_F = 0$ (bare BEC) in **a** and $\Omega_F = 1.3\ E_r$ (SO-coupled BEC) in **b**, followed by various hold times ($t_{hold}$) in the dipole trap. The TOF images labeled by $t_{hold} = -1$ ms are taken right before the application of $E_\sigma$. The bare spin components (labeled by $m_F$, with $|\downarrow\rangle$ in red and $|\uparrow\rangle$ in blue) are separated along the horizontal axis. The vertical axis shows the atoms' mechanical momentum $\hbar k$ along the SOC direction ($\hat{y}$). The color scale reflects the measured optical density (OD, see Methods). The total condensate atom number of the initial state at $\Omega_I$ is $N_c \sim (1-2) \times 10^4$ with trap frequencies $\omega_z \sim 2\pi \times (37 \pm 5)$ Hz and $\omega_x \sim \omega_y \sim 2\pi \times (205 \pm 15)$ Hz. The TOF images (and associated analyzed quantities presented later) are typically the average of a few repetitive measurements

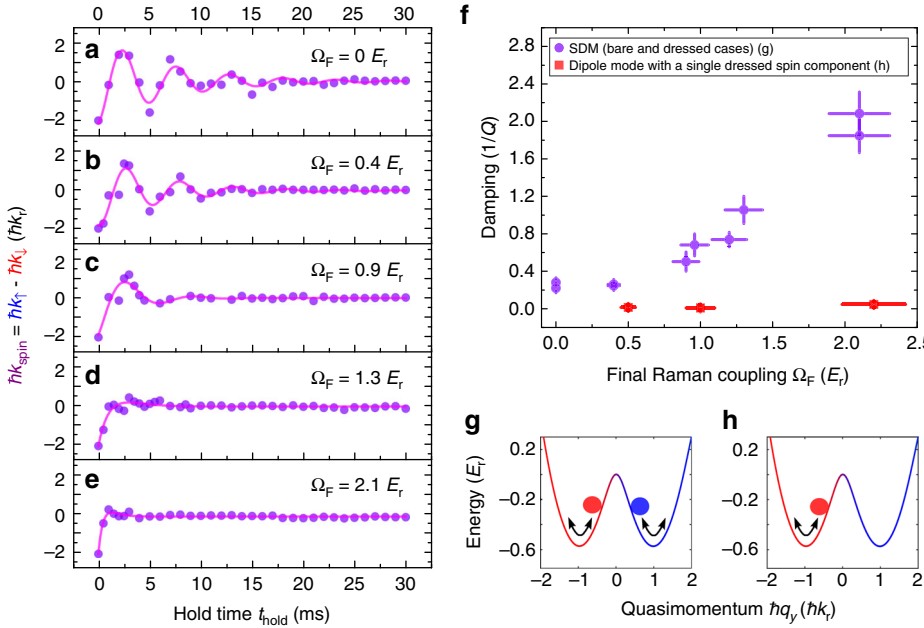

**Fig. 3** Momentum damping at different $\Omega_F$, for SDM and the dipole mode of a single dressed spin component. **a–e** Relative momentum oscillations in SDM, $\hbar k_{spin}$, as a function of $t_{hold}$ at various $\Omega_F$. The experimental data (scatters) are fitted to a damped sinusoidal function (line) to extract the inverse quality factor $1/Q$ of the oscillations. **f** Momentum damping (quantified by $1/Q$) versus $\Omega_F$. The error bar of $1/Q$ is the standard error of the fit. The purple circle data correspond to the SDM (illustrated by **g**) and the red square data correspond to the dipole mode of a BEC with a single dressed spin component prepared in $|\downarrow'\rangle$ (illustrated by **h**). In **g**, **h**, the representative band structure is calculated at $\Omega = 1.0\ E_r$

by the inverse quality factor $1/Q = t_{trap}/(\pi\tau_{damp})$, where $1/t_{trap}$ is the trap frequency along $\hat{y}$ taking into account of the effective mass for the dressed case (see Methods). We observe that the damping ($1/Q$) is higher for larger $\Omega_F$, summarized by the purple data in Fig. 3f. Additionally, we have performed two control experiments, which suggest that SOC alone cannot cause momentum damping and thermalization if there are no collisions between the two dressed spin components. Only when there is SDM would notable thermalization be observed within the time of measurement. First, we measure the dipole oscillations[17,22] of a SO-coupled BEC with a single dressed spin component prepared in $|\downarrow'\rangle$ at various $\Omega_F$. This gives a spin current as well as a net mass current. We observe (e.g., Supplementary Fig. 1 in Supplementary Note 1) that these single-component cases exhibit very small damping ($1/Q < 0.05$, summarized by the red square data in Fig. 3f) and negligible thermalization. In another control experiment, we generate only an AC mass current without a spin current by exciting in-phase dipole oscillations of two dressed spin components of a SO-coupled BEC without relative collisions (SDM). This experiment also reveals very small damping and negligible thermalization (see Supplementary Fig. 2 in Supplementary Note 1).

**Thermalization and spin current**. We now turn our attention to the thermalization, i.e., the reduction of condensate fraction due to collisions between the two spin components. To quantitatively describe the observed thermalization, the integrated optical density of the atomic cloud in each spin component is fitted to a 1D bimodal distribution to extract the total condensate fraction $f_c = N_c/N$ (see Methods) with $N$ being the total atom number and $N_c$ the total condensate atom number (including both spin states). The time ($t_{hold}$) evolution of the measured $f_c$ is plotted for the bare ($\Omega_F = 0$) and dressed ($\Omega_F = 1.3\ E_r$ and $2.1\ E_r$) cases in Fig. 4a. In all the cases, we observe that $f_c$ first decreases with time before it no longer changes substantially (within the experimental uncertainty) after some characteristic thermalization time

($\tau_{therm}$). To capture the overall behavior of the thermalization, we fit the smoothed $t_{hold}$-dependent data of $f_c$ to a shifted exponential decay $f_c(t_{hold}) = f_s + (f_i - f_s)\exp(-t_{hold}/\tau_{therm})$, where $\tau_{therm}$ represents the time constant for the saturation of the decreasing condensate fraction and $f_s$ the saturation condensate fraction (see Methods). We obtain $\tau_{therm} = 3.8(4)$ ms, $2.4(3)$ ms, and $0.4(1)$ ms for $\Omega_F = 0$, $1.3\ E_r$, and $2.1\ E_r$, respectively. Besides, a notably larger condensate fraction ($f_s$) is left for a larger $\Omega_F$, where $f_s \sim 0.2$, $0.3$, and $0.4$ for $\Omega_F = 0$, $1.3\ E_r$, and $2.1\ E_r$, respectively. Since thermalization during our measurement time is induced by the SDM, the observation that a larger $\Omega_F$ gives rise to a smaller $\tau_{therm}$ and a larger $f_s$ (Fig. 4b) thus less thermalization is understood as due to the stronger SDM damping (smaller $\tau_{damp}$) at larger $\Omega_F$, stopping the relative collision between the two spin components thus the collision-induced thermalization earlier.

The coherent spin current is phenomenologically defined as $I_s = I_\uparrow - I_\downarrow$ (see Methods), where $I_{\sigma=\uparrow,\downarrow}$ is given by:

$$I_\sigma = \frac{N_c^\sigma}{L^\sigma}\nu^\sigma = f_c^\sigma \nu^\sigma \frac{N^\sigma}{L^\sigma} \qquad (2)$$

Here, $\sigma$ labels the physical quantities associated with the spin component $\sigma$, $L^\sigma$ is the in situ BEC size along the current direction, and $\nu^\sigma = \hbar k_\sigma/m$. We exclude the contribution from the thermal atoms as only the condensate atoms participate in the coherent spin transport. In our experiments, $N^\uparrow/L^\uparrow \approx N^\downarrow/L^\downarrow$ is not observed to decrease significantly with $t_{hold}$, and $f_c^\uparrow \approx f_c^\downarrow \approx f_c$, thus the relaxation of $I_s$ is mainly controlled by that of $f_c^\uparrow \nu^\uparrow - f_c^\downarrow \nu^\downarrow \approx f_c(\nu^\uparrow - \nu^\downarrow)$. Therefore, the SDM damping (reduction of $\nu^\uparrow - \nu^\downarrow$) and thermalization (reduction of $f_c$) provide the two main mechanisms for the relaxation of coherent spin current.

Figure 4c shows the normalized $I_s$ as a function of $t_{hold}$ extracted (see Methods) for $\Omega_F = 0$ and $1.3\ E_r$. In the bare case, the spin current oscillates around and decays to zero. In the dressed case, the spin current relaxes much faster to zero without completing one oscillation. Fitting $I_s$ versus $t_{hold}$ to a damped

sinusoidal function for $\Omega_F = 0$ or to an exponential decay for $\Omega_F = 1.3\ E_r$ (with no observable $I_s$ oscillations) allows us to extract the spin current decay time constant $\tau_{spin}$, which is 5.1(8) ms and 0.5(0) ms, respectively. In the dressed case $I_s$ decays much faster compared to the bare case because both $\tau_{damp}$ and $\tau_{therm}$ are much smaller due to stronger SDM damping. In the bare case, the thermalization plays a more important role in the relaxation of $I_s$ due to the larger reduction of condensate fraction $(f_i - f_s)$ compared to the dressed case.

**Observation of deformed atomic clouds and BEC shape oscillations**. In addition to the SDM damping and thermalization, the atomic clouds can exhibit other rich dynamics after the application of $E_\sigma$. We observe deformation of atomic clouds at early stages of the SDM, as shown in Fig. 5a–d. Figure 5b, d shows the observation of an elongated atomic cloud at $t_{hold} = 0.5$ ms in the dressed case at $\Omega_F = 2.1\ E_r$, in comparison with the atomic cloud

at $t_{hold} = 0.5$ ms in the bare case shown in Fig. 5a, c. Figure 5c, d shows the integrated optical density (denoted by $OD_y$) of the atomic cloud versus the $y$ direction, obtained by integrating the measured optical density over the horizontal direction in TOF images. The momentum distribution of the atoms at $\Omega_F = 2.1\ E_r$ has lower $OD_y$ and is more elongated without a sharp peak along the SOC direction, in comparison with the bare case that has higher $OD_y$ and a more prominent peak momentum. Furthermore, we observe that the relaxation of the spin current is accompanied by BEC shape oscillations[52–54] (Fig. 5e, f), which remain even after the spin current is fully damped. These additional experimental observations are closely related to the spin current relaxation, as discussed below.

**GPE simulations and interpretations**. We have performed numerical simulations for the SDM based on the 3D time-dependent Gross-Pitaevskii equation (GPE), using similar

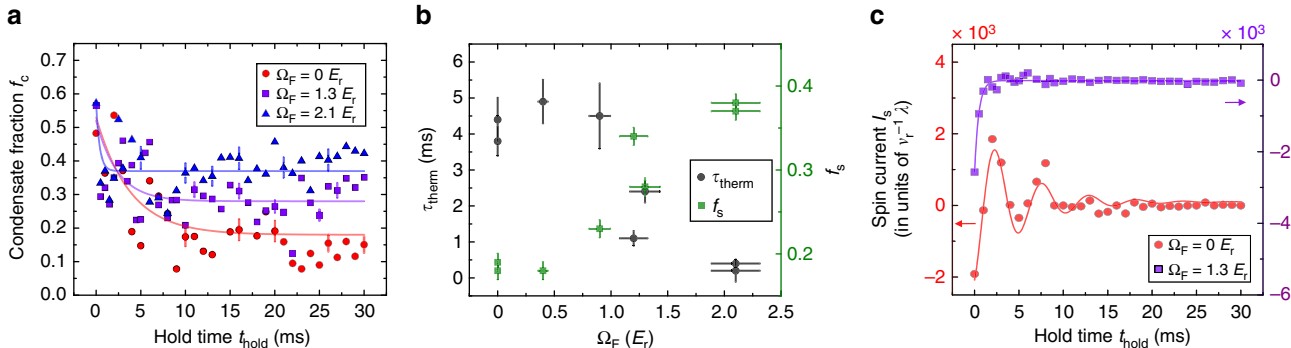

**Fig. 4** Thermalization and spin current. **a** The measured condensate fraction $f_c = N_c/N$ as a function of $t_{hold}$ for SDM in the bare case (no SOC, $\Omega_F = 0$) and the dressed cases (with SOC, $\Omega_F = 1.3\ E_r$ and $\Omega_F = 2.1\ E_r$). Representative error bars show the average percentage of the standard error of the mean. The solid curves are the shifted exponential fits to the smoothed $f_c$ (see Methods). The initial condensate fraction (not shown) at $\Omega_I$ (measured at $t_{hold} = -1$ ms) is ~0.6–0.7 for all the cases. **b** The saturation time constant $\tau_{therm}$ of the decreasing $f_c$ and the saturation condensate fraction $f_s$ versus $\Omega_F$, where the vertical error bar is the standard error of the fit. **c** Spin current $I_s$ (normalized by $v_r/\lambda = 7.4 \times 10^3/s$, where $v_r \sim 6$ mm/s is the recoil velocity) as a function of $t_{hold}$ for $\Omega_F = 0$ and $1.3\ E_r$. The solid curves are fits (see text)

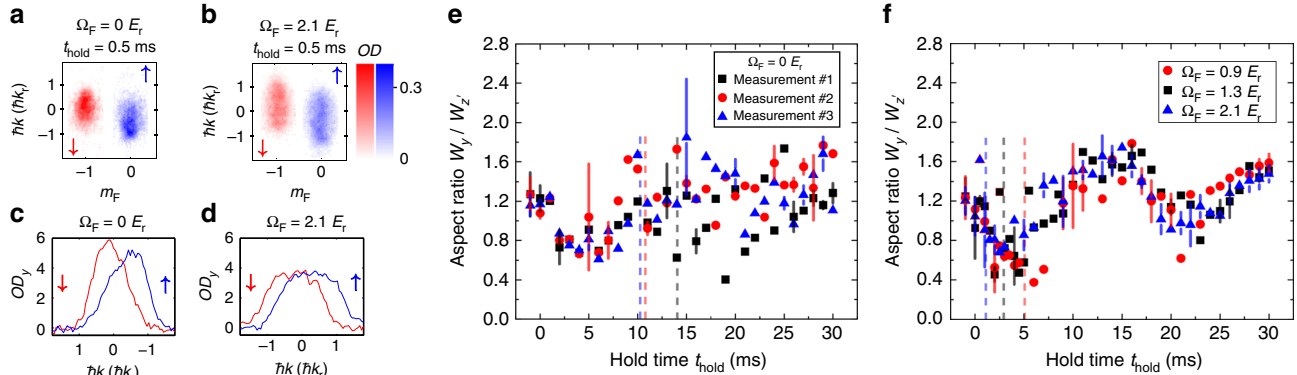

**Fig. 5** Observation of deformed atomic clouds and BEC shape oscillations. **a–d** Observation of deformed atomic clouds at early stages of the SDM. **a**, **b** TOF images for $\Omega_F = 0$ and $\Omega_F = 2.1\ E_r$ at $t_{hold} = 0.5$ ms are shown for comparison. The corresponding integrated optical density $(OD_y)$ versus the momentum in the SOC direction $(\hat{y})$ for the spin down and up components is shown respectively in **c**, **d**. **e**, **f** Observation of BEC shape oscillations. The data showing the aspect ratio $W_y/W_{z'}$ (see Methods) of the condensate measured at various $t_{hold}$ are extracted from the SDM measurements in Fig. 3, except for the additional measurements #2 and #3 in **e**. **e** For the three independent measurements in the bare case, the observed oscillations possess a complicated behavior without having a well-defined frequency given the error bars and the fluctuation in the data. Select TOF images for measurement #1 are shown in Fig. 2a. **f** In the dressed cases, aspect ratio oscillations with a well-defined frequency are observed in measurements at three different $\Omega_F$. The average frequency of the three aspect ratio oscillations obtained from the damped sinusoidal fit is around 58 Hz, consistent with the expected frequency for the $m = 0$ quadrupole mode $f_{m=0} = \sqrt{2.5}\omega_z/(2\pi) \sim 59$ Hz for a cigar-shape BEC in the limit of $\omega_z/\omega_{x,y} \ll 1$[52]. Note that $\omega_z$ is not modified by Raman lasers and thus does not depend on $\Omega_F$. Select TOF images for $\Omega_F = 1.3\ E_r$ are shown in Fig. 2b. The representative error bars in (**e**, **f**) are standard deviation of at least three measurements. The dashed lines indicate $t_{hold} \sim 2\tau_{damp}$ after which the SDM is fully damped out

parameters as in the experiments. The $\Omega_F$-dependent $1/Q$ extracted from the GPE-simulated SDM (Fig. 6a–c) shows qualitative agreement with the experimental measurements (Fig. 6d, e). Quantitatively, we notice that the GPE simulation generally underestimates the momentum damping compared to the experimental observation (Fig. 6e), especially at low $\Omega_F$ (including the bare case). This is possibly related to the fact that our GPE simulation cannot treat thermalization (which is more prominent at low $\Omega_F$) and effects of thermal atoms. Nonetheless, the in situ (real space) spin-dependent density profiles (Fig. 6f–j) of the BECs calculated from the GPE simulations have provided

important insights to understand why SOC can significantly enhance the SDM damping. Figure 6f shows that the initial BEC (just before applying $E_\sigma$) in the trap is in an equal superposition of bare spin up and down states. Figure 6g–j shows the density profiles of the BECs at $t_{hold} = 1.5$ ms (after applying $E_\sigma$ in the trap with four different $\Omega_F$ (see Supplementary Movies 2, 4 and 5 in Supplementary Note 3). For the bare case, the two spin components fully separate in the real space within the trap. As $\Omega_F$ becomes larger, we observe that only a smaller portion of atoms in each spin component is well separated, as marked by the white arrows. Concomitantly, a larger portion of atoms appears to get

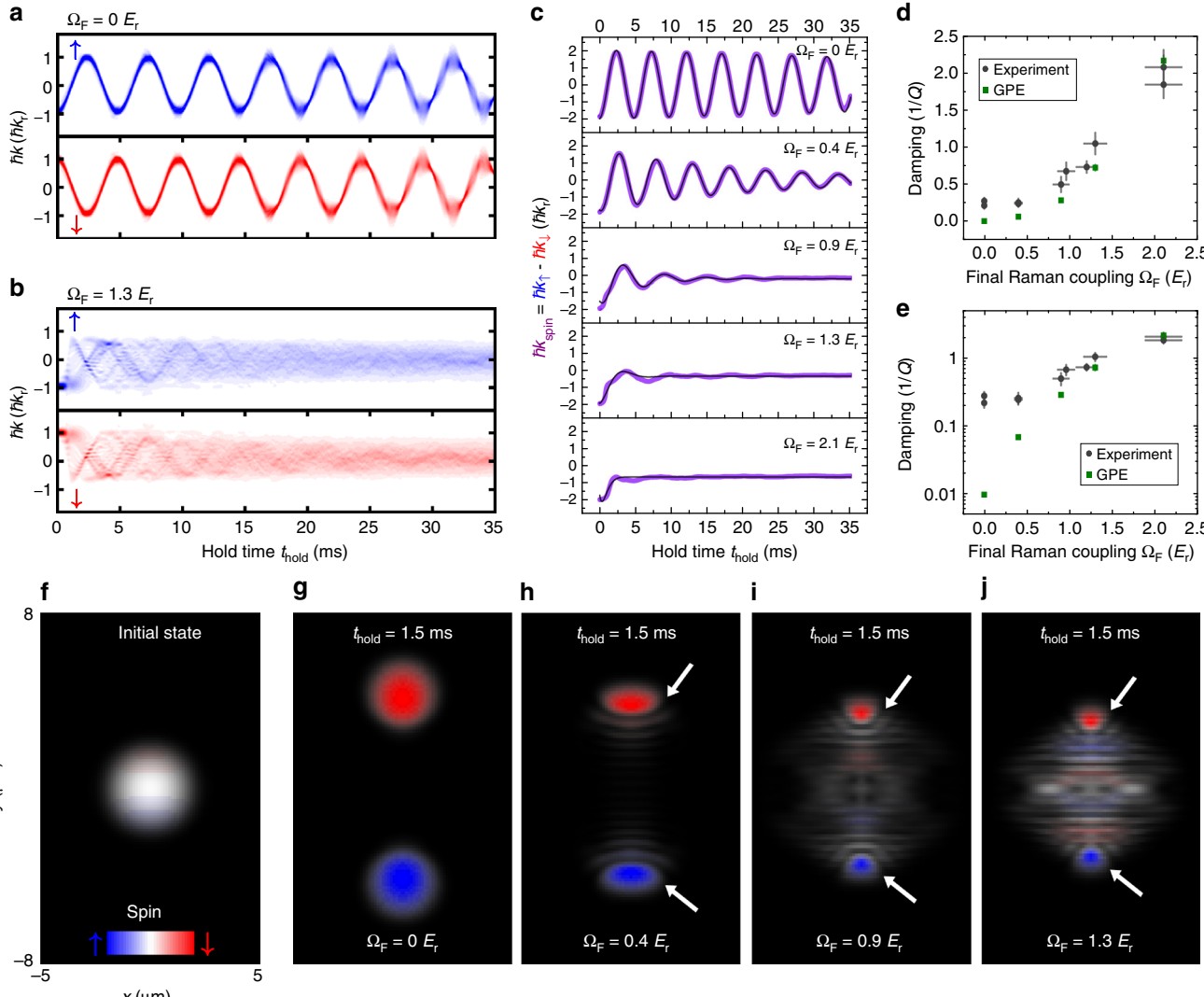

**Fig. 6** GPE simulated SDM at various $\Omega_F$ and the extracted SDM damping compared with experiment. **a, b** GPE simulations of the 1D momentum-space density distributions of the two bare spin components as a function of $t_{hold}$ for the SDM at $\Omega_F = 0$ and $\Omega_F = 1.3\ E_r$, respectively. The 1D momentum density $\rho_\sigma(k_y)$ is obtained by integrating the 3D momentum density along $k_x$ and $k_z$, i.e., $\rho_\sigma(k_y) = \int \rho_\sigma(k_x, k_y, k_z)dk_x dk_z$. Then, these integrated 1D atomic momentum densities for sequential hold times ($t_{hold}$) are combined to show the atomic density in momentum space along the SOC direction versus $t_{hold}$. **c** GPE simulations of the SDM damping versus $t_{hold}$ at various $\Omega_F$. The violet lines are the $\hbar k_{spin}$ (defined as the difference between the CoM momenta of the two spin components) as a function of $t_{hold}$ for various $\Omega_F$. The CoM momentum ($\hbar k_{\uparrow,\downarrow}$) of each bare spin component (at a given $t_{hold}$) is calculated by taking a density-weighted average of the corresponding 1D momentum density distributions such as those shown in **a, b**. The black lines are damped sinusoidal fits for the calculated $\hbar k_{spin}$ to extract the corresponding SDM damping ($1/Q$) which is shown in **d** along with the experimental data reproduced from Fig. 3f. **e** Replotting of **d** with $1/Q$ shown in logarithmic scale. **f–j** In situ (real space) atomic densities calculated from GPE simulations. **f** Initial in situ 2D density at $\Omega = \Omega_I$ (right before applying spin-dependent electric fields $E_\sigma$). **g–j** In situ 2D density at $t_{hold} = 1.5$ ms (after the application of $E_\sigma$) for $\Omega_F = 0$, 0.4 $E_r$, 0.9 $E_r$, and 1.3 $E_r$, respectively. For **f–j**, the density is designated by brightness and the bare spin polarization by colors (red: $\downarrow$, blue: $\uparrow$, white: equal spin populations). The 2D densities $\rho_\sigma(x, y)$ in **f–j** are obtained by integrating the 3D atomic density along $z$, i.e., $\rho_\sigma(x, y) = \int \rho_\sigma(x, y, z)dz$. In this figure, the simulations used the following parameters representative of our experiment: $\Omega_I = 5.2\ E_r$, $\delta_R = 0$, $N_c = 1.6 \times 10^4$, $\omega_z = 2\pi \times 37$ Hz, $\omega_x = \omega_y = 2\pi \times 205$ Hz, $t_E = 1.0$ ms

stuck around the trap center and form a prominent standing wave pattern, which we interpret as density modulations arising from the interference between the BEC wavefunctions of the two dressed spin components when $|\uparrow\,'\rangle$ and $|\downarrow\,'\rangle$ are no longer orthogonal in the presence of SOC (see Fig. 7a)[21,55–58]. Compared to the bare case, the formation of density modulations in the dressed case can lead to more deformed clouds in both the real and momentum spaces at early stages in the SDM, as revealed by the GPE simulations (Fig. 6a, b, f–j; Supplementary Movies 2, 4 and 5 in Supplementary Note 3). This is consistent with our experimental observation of a highly elongated momentum distribution of the atomic cloud along the SOC direction $(\hat{y})$ at early instants in the SDM of a SO-coupled BEC (Fig. 5b, d).

In addition to density modulations, our GPE simulation also reveals complex spatial modulation in the phase of the BEC wavefunctions (see Supplementary Fig. 9 and Supplementary Movies 3 and 6 in Supplementary Note 3). Such distortions of BEC wavefunctions in the amplitude (which determines the density) and the phase contribute to quantum pressure[59] and local current kinetic energy (see Methods) respectively, two forms of the kinetic energy that do not contribute to the global

translational motion (or CoM kinetic energy) of each spin component. The sum of the CoM kinetic energy, quantum pressure, and local current kinetic energy is the total kinetic energy (see Methods). We have used GPE to calculate the time evolution of these different parts of kinetic energy for the dressed case, showing that the damping of the CoM kinetic energy (which decays to zero at later times) is accompanied by (thus likely related to) prominent increase of the quantum pressure and the local current kinetic energy (both remain at some notable finite values at later times) (see Fig. 8e–h). The increasing quantum pressure and local current kinetic energy may reflect the emergence of excitations that do not have the CoM kinetic energy. This is consistent with the experimentally observed generation of BEC shape oscillations (Fig. 5e, f), whose kinetic energy can be accounted for by the quantum pressure and the local current kinetic energy. Note that the excitation of BEC shape oscillations may also be understood by the observation of deformed clouds at early stages of the SDM (Fig. 5a–d), because the deformed shape of the BEC is no longer in equilibrium with the trap and thus initiates the shape oscillations. The observed BEC shape oscillations remain even after the SDM is completely damped in both bare and dressed cases. This indicates that the

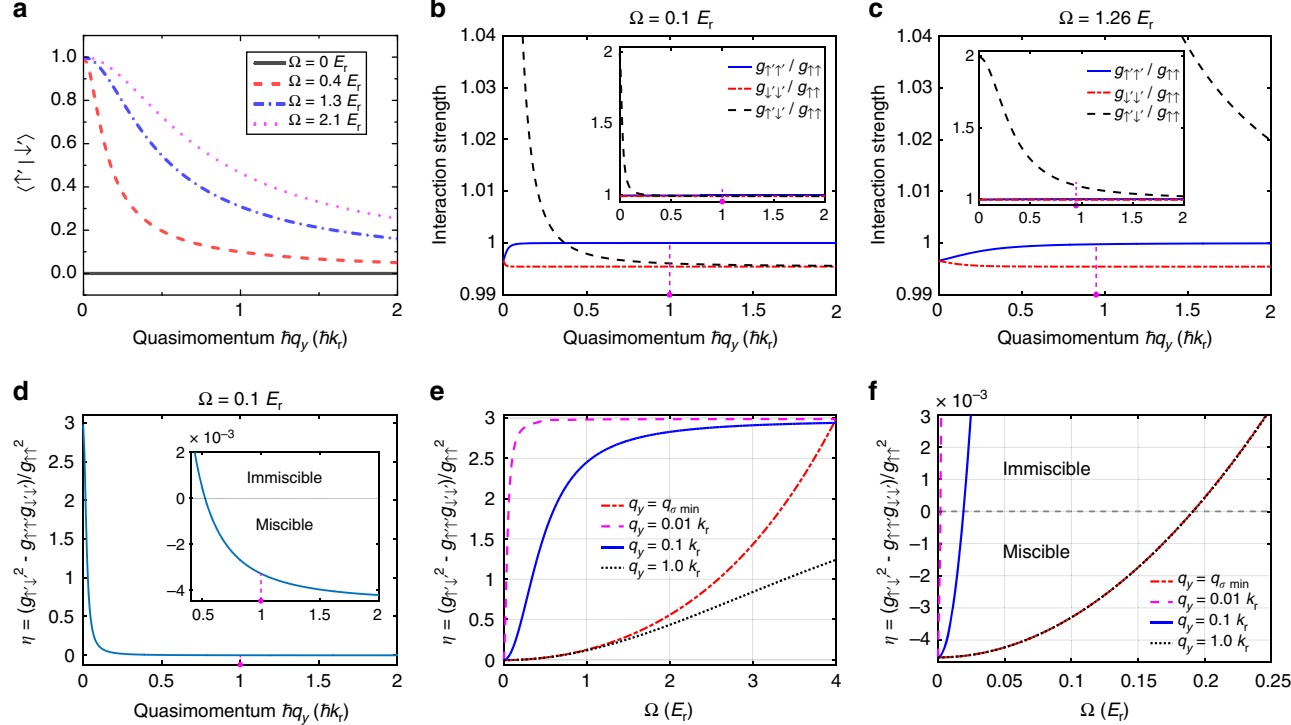

**Fig. 7** Calculated nonorthogonality, effective interaction parameters, and immiscibility for two dressed spin states. In (**a**–**f**), the calculations consider $|\uparrow\,'\rangle$ and $|\downarrow\,'\rangle$ located respectively at $\hbar q_y$ and $-\hbar q_y$. **a** When $\Omega = 0$, the nonorthogonality is zero because the two bare spin components are orthogonal. When $\Omega \neq 0$, either increasing $\Omega$ or decreasing $q_y$ would increase $\langle\uparrow'|\downarrow'\rangle$, giving rise to stronger interference and more significant density modulations in the spatially overlapped region of the two dressed spin components. **b**, **c** Effective interspecies ($g_{\uparrow'\downarrow'}$) and intraspecies ($g_{\uparrow'\uparrow'}$, $g_{\downarrow'\downarrow'}$) interaction parameters versus quasimomentum at $\Omega = 0.1\,E_r$ and $1.26\,E_r$, respectively. When $\Omega$ increases or $q_y$ decreases, $g_{\uparrow'\downarrow'}$ increases while $g_{\uparrow'\uparrow'}$ and $g_{\downarrow'\downarrow'}$ almost remain at the bare values. As $q_y \to 0$ at any finite $\Omega$, $g_{\uparrow'\downarrow'} \to 2g_{\uparrow'\uparrow'}$ or $2g_{\downarrow'\downarrow'}$, which is the upper bound of $g_{\uparrow'\downarrow'}$ (see Methods). The inset of **b**, **c** zooms out to show the maximum. **d** shows the immiscibility metric $\eta = \left(g_{\uparrow'\downarrow'}^2 - g_{\uparrow'\uparrow'}g_{\downarrow'\downarrow'}\right)/g_{\uparrow\uparrow}^2$ in Eq. (13) (see Methods) versus $\hbar q_y$ corresponding to **b**. $\eta < 0$ means miscible, and $\eta > 0$ means immiscible. Over the range of plotted $\hbar q_y$, **d** can be miscible or immiscible depending on $\hbar q_y$. The inset of **d** zooms in to focus on the sign change of $\eta$. The vertical dotted line in (**b**–**d**) indicates $\hbar q_{\sigma\,\mathrm{min}}$ corresponding to the $\Omega$ in each case. The calculations are performed in the two-state picture described by Eq. (1) with $\delta_R = 0$. **e**, **f** Immiscibility metric $\eta$ versus $\Omega$ for various $q_y$. In **e**, as $\Omega$ becomes larger or $q_y$ becomes smaller, the two dressed spin components can become more immiscible until $\eta$ reaches the maximum value set by the upper bound of $g_{\uparrow'\downarrow'}$ (see also **b**, **c**). **f** Zoom-in of **e** showing the miscible to immiscible transition (indicated by the gray dashed line at $\eta = 0$) as a function of $\Omega$ for various $q_y$. The red dot-dashed line corresponds to two dressed spin components located respectively at the band minima $q_{\sigma\,\mathrm{min}}$, showing the well-known miscible to immiscible transition around $0.2\,E_r$ for a stationary SO-coupled BEC. In the dynamical case studied here, BECs can be located away from the band minima and approach $q_y = 0$, becoming immiscible even when $\Omega < 0.2\,E_r$ for small enough $q_y$

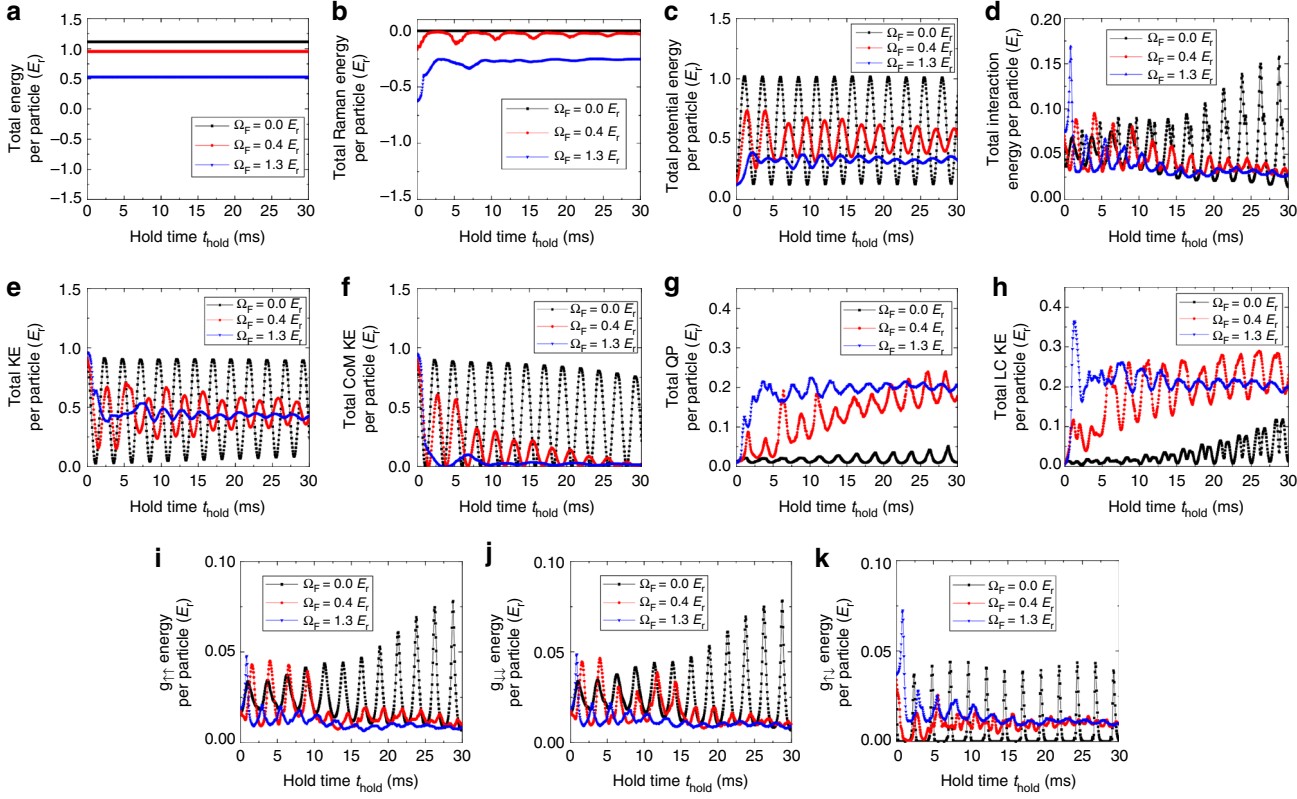

**Fig. 8** Time ($t_{hold}$) evolution of different forms of energies per particle at different $\Omega_F$ as calculated by GPE. **a** The total energy is the sum of the total Raman energy, total potential energy, total interaction energy, and the total KE. The result in **a** confirms that the total energy is conserved during $t_{hold}$. **b** Total Raman energy. **c** Total potential energy. **d** Total interaction energy, sum of the bare interaction energies in **i**–**k**. **e** Total KE, sum of different types of kinetic energies in **f**–**h**. **f** Total CoM KE. **g** Total QP. **h** Total LC KE. **i** $g_{\uparrow\uparrow}$ interaction energy. **j** $g_{\downarrow\downarrow}$ interaction energy. **k** $g_{\uparrow\downarrow}$ interaction energy

BECs are still excited even after the CoM relaxes to the single-particle band minima within the time of measurement.

## Discussion

Previous studies in stationary SO-coupled BECs (located at ground dressed band minima) have found that increasing $\Omega$ drives a miscible to immiscible phase transition at $\Omega \sim 0.2\ E_r$ due to the increased effective interspecies interaction (characterized by the interaction parameter $g_{\uparrow'\downarrow'}$)[21,55–57,60]. In the miscible phase, the two dressed spin components have substantial spatial overlap, where density modulations form. It is important to note that the effective interactions, immiscibility and interference between the two dressed spin components depend on the quasimomentum ($\hbar q_y$) and $\Omega_F$ (Fig. 7, see Methods for details). Therefore, in the dynamical case studied here, these properties vary with time and can be notably different from those in the stationary case. During the SDM, the two dressed spin components are forced to collide due to $E_\sigma$. This can give rise to interference-induced density modulations in their spatially overlapped region even when they are immiscible. In addition, the BECs during the SDM can be located away from the band minima and approach $q_y = 0$. For the two dressed spin components with quasimomenta $\pm\hbar q_y$, either increasing $\Omega_F$ or decreasing $|q_y|$ (towards 0) would increase $\langle\uparrow'|\downarrow'\rangle$ (Fig. 7a), giving rise to stronger interference and more significant density modulations. Such increased non-orthogonality between the two dressed spin states also notably increases the effective interspecies interaction ($g_{\uparrow'\downarrow'}$) to become even larger than the effective intraspecies interactions ($g_{\uparrow'\uparrow'} \approx g_{\downarrow'\downarrow'}$) (Fig. 7b, c),

enhancing further the immiscibility (Fig. 7d–f). For example, Fig. 7d shows the calculated immiscibility metric (see Methods), $\eta = (g_{\uparrow'\downarrow'}^2 - g_{\uparrow'\uparrow'}g_{\downarrow'\downarrow'})/g_{\uparrow'\uparrow'}^2$, versus $\hbar q_y$ corresponding to Fig. 7b. Notice that when $\Omega$ is large enough, $|\uparrow'\rangle$ and $|\downarrow'\rangle$ can become immiscible in the whole range of quasimomentum that a BEC can access during the SDM. Figure 7e shows $\eta$ versus $\Omega$ at various $\hbar q_y$. We see that as $\Omega$ becomes larger or $q_y$ becomes smaller, the two dressed spin components can become more immiscible (i.e., $\eta$ becomes more positive) until $\eta$ reaches the maximum value set by the upper bound of $g_{\uparrow'\downarrow'}$. Figure 7f zooms in the region of small $\Omega$ in Fig. 7e to focus on the sign change of $\eta$ from negative to positive, which indicates the miscible to immiscible transition. Note that the red dot-dashed line (for $q_y = q_{\sigma\ min}$) corresponds to two dressed spin components located respectively at the band minima $q_{\sigma\ min}$, showing the well-known miscible to immiscible transition around $0.2\ E_r$ for a stationary SO-coupled BEC. In the dynamical case studied here, BECs can be located away from the band minima and approach $q_y = 0$, becoming immiscible even when $\Omega < 0.2\ E_r$ for small enough $q_y$.

We have performed several additional control GPE simulations, showing that the presence or the enhancement of any of these three factors can increase the damping of the relative motion between two colliding BECs: (1) interference (Supplementary Fig. 5 and Supplementary Movie 1 in Supplementary Note 3), (2) immiscibility (Supplementary Fig. 4 and Supplementary Table 1 in Supplementary Note 3), and (3) interactions (Supplementary Figs. 4, 6, 7 and 8 and Supplementary Table 1 in Supplementary Note 3), presumably by distorting the BEC wavefunctions (see Supplementary Movies 1–6 in Supplementary Note 3) irreversibly in the presence of interactions to decrease the

CoM kinetic energy while increasing the quantum pressure and the local current kinetic energy. Therefore, enhanced immiscibility, interference, and interactions can all increase the damping of the SDM. For simulations in the absence of interactions, we do not observe irreversible damping within the simulation time of 100 ms (Supplementary Figs. 6–8 in Supplementary Note 3), suggesting that the interactions play an essential role for the damping mechanisms.

The physical mechanisms and processes revealed in our work may provide insights to understand spin transport in interacting SO-coupled systems. Our experiment also provides an exemplary study of the evolution of a quantum many-body system, including the generation and decay of collective excitations, following a non-adiabatic parameter change (quench). Such quench dynamics has been of great interest to study many outstanding questions in many-body quantum systems. For example, how such a system, initially prepared in the ground state but driven out of equilibrium due to a parameter quench that drives the system across a quantum phase transition, would evolve to the new ground state or thermalize has attracted great interests (see, e.g., a recent study where coherent inflationary dynamics has been observed for BECs crossing a ferromagnetic quantum critical point[61]). In our case, the sudden reduction of $\Omega$ in the Hamiltonian Eq. (1) excites the coherent spin current, whose relaxation is strongly affected by SOC and is related to the SDM damping as well as thermalization. Besides, the relaxation may be accompanied by the generation of other collective excitations such as BEC shape oscillations. Furthermore, compared to the bare case, the SOC-enhanced damping of the SDM notably reduces the collision-induced thermalization of the BEC, resulting in a higher condensate fraction left in the BEC. This condensate part exhibits a more rapid localization of its CoM motion, which may be more effectively converted to other types of excitations (associated with the SOC-enhanced distortion of the BEC wavefunctions). These features suggest that SOC opens pathways for our interacting quantum system to evolve that are absent without interactions, in our case providing new mechanisms for the spin current relaxation. Experiments on SO-coupled BECs, where many parameters can be well controlled in real time and with the potential of adding other types of synthetic gauge fields, may offer rich opportunities to study nonequilibrium quantum dynamics[62], such as Kibble–Zurek physics while quenching through quantum phase transitions[63], and superfluidity[16,33] in SO-coupled systems.

## Methods

**Spin-dependent vector potentials.** In Eq. (1), the eigenenergies at $\delta_R = 0$ are given by:

$$E_\pm\left(q_y\right) = \frac{\hbar^2 q_y^2}{2m} + E_r \pm \sqrt{\left(\frac{\Omega}{2}\right)^2 + \left(\frac{\hbar^2 k_r q_y}{m}\right)^2} \quad (3)$$

For $\Omega < \Omega_c$, the ground band of the energy-quasimomentum dispersion has two minima at:

$$q_{\sigma\,\min}(\Omega) = \pm k_r \sqrt{1 - \left(\Omega/\Omega_c\right)^2} \quad (4)$$

The state of the atoms associated with each minimum at $q_{\sigma\,\min}$ can be regarded as a dressed spin state. For a double minima band structure, we thus have two dressed spin components $\sigma = |\downarrow\,'\rangle$ and $|\uparrow\,'\rangle$ that constitute a pseudo spin-1/2 system (when $\Omega = 0$, $|\uparrow\,'\rangle$ and $|\downarrow\,'\rangle$ become the bare spin $|\uparrow\rangle$ and $|\downarrow\rangle$, respectively). The energy spectrum expanded around each $q_{\sigma\,\min}$ as $E(q_y) = \hbar^2(q_y - q_{\sigma\,\min})^2/(2m^*)$ is analogous to the Hamiltonian describing a charged particle with charge $Q$ in a magnetic vector potential $A$, $\hat{H} = (\hat{p}_y - QA)^2/(2m_Q)$, where $m^*$ is the effective mass of a dressed atom and $m_Q$ is the mass of the charged particle. Therefore, we can identify the quasimomentum ($\hbar q_y$) with the canonical momentum ($\hat{p}_y = -i\hbar \frac{\partial}{\partial y}$), and $\hbar q_{\sigma\,\min}$ with the light-induced spin-dependent vector potentials ($A_\sigma$, by setting $Q = 1$ for our case[20]). The velocity operator corresponding to the mechanical momentum,

$\hat{v}_y = -[\hat{H}, y]/(i\hbar) = (\hat{p}_y - QA)/m_Q$, thus corresponds to $\hbar(q_y - q_{\sigma\,\min})/m^*$. These spin-dependent vector potentials $A_\sigma$ (represented by $\hbar q_{\sigma\,\min}$) are tunable by $\Omega$. For example, as seen in Fig. 1c, we can decrease $\Omega$ to separate the two $\hbar q_{\sigma\,\min}$ or increase $\Omega$ to combine them in the quasimomentum space.

**Effects of the neglected $|m_F = +1\rangle$ state.** We apply an external bias magnetic field $\mathbf{B} = B\hat{z}$ (~5 gauss) to Zeeman split the energies $E_{-1}$, $E_0$, and $E_{+1}$ of the $|m_F = -1\rangle$, $|m_F = 0\rangle$, and $|m_F = +1\rangle$ sublevels respectively (in the $F = 1$ hyperfine state of $^{87}$Rb atoms), where $E_{-1} - E_0 = \hbar\omega_Z$, $E_0 - E_{+1} = \hbar\omega_Z - 2\varepsilon$, $\hbar$ is the reduced Planck constant and $\varepsilon = (E_{-1} + E_{+1})/2 - E_0$ is the quadratic Zeeman shift. The frequency difference between the two Raman lasers is $\Delta\omega_R/(2\pi) = 3.5$ MHz. The Raman detuning $\delta_R = \hbar(\Delta\omega_R - \omega_Z)$ is controlled by $B$ that controls $\hbar\omega_Z$. In a first-order approximation, the third state $|m_F = +1, \hbar k = \hbar(q_y - 3k_r)\rangle$ can be excluded in Eq. (1) due to the quadratic Zeeman shift ($2\varepsilon \sim 0.9\,E_r$) from $B$ but can be included in the following three-state Hamiltonian:

$$H_3 = \begin{pmatrix} \frac{\hbar^2}{2m}\left(q_y + k_r\right)^2 - \delta_R & \frac{\Omega}{2} & 0 \\ \frac{\Omega}{2} & \frac{\hbar^2}{2m}\left(q_y - k_r\right)^2 & \frac{\Omega}{2} \\ 0 & \frac{\Omega}{2} & \frac{\hbar^2}{2m}\left(q_y - 3k_r\right)^2 + \delta_R + 2\varepsilon \end{pmatrix} \quad (5)$$

In our SDM experiments, we always maintain approximately equal spin populations in the $|\downarrow\rangle = |m_F = -1\rangle$ and $|\uparrow\rangle = |m_F = 0\rangle$ states both in the initial dressed state prepared at $\Omega_I$ and in the final dressed state at $\Omega_F$ (with approximately equal populations also achieved in $|\downarrow\,'\rangle$ and $|\uparrow\,'\rangle$ at $\Omega_F$). In Eq. (1) based on the two-state picture in the main text, $\delta_R = 0$ can give rise to such balanced (dressed/bare) spin populations at any given $\Omega$. However, in Eq. (5) with $\delta_R = 0$, a finite $\Omega$ can lead to unbalanced (dressed/bare) spin populations. Therefore, in our experiment $\delta_R$ at a given $\Omega$ has to be changed to $\delta'(\Omega, \varepsilon)$ to achieve the balanced spin populations (note that in the double minima regime of Eq. (5), this requirement is in a good approximation equivalent to the so-called balanced band condition where the two minima in the ground dressed band have equal energy). Such an effect is addressed in details in ref. [21]. In our case, also note that including the third state in Eq. (5) would cause the actual transition from the double minima to single minimum to occur at $\Omega_c \sim 4.7\,E_r$ rather than at $\Omega_c = 4.0\,E_r$ as would be predicted by Eq. (1). Additionally, Eq. (5) is used for plotting Figs. 1c and 3g, h, which more precisely means $\delta_R = \delta'(\Omega, \varepsilon)$ to achieve the balanced spin populations for the corresponding $\Omega$. In the following, we use Eq. (5) to describe the initial state preparation process.

**Initial state preparation, spin population balance, and imaging process.** We create spin-polarized $^{87}$Rb BECs in $|m_F = 0\rangle$ in an optical dipole trap consisting of three cross laser beams (with a third beam added to the double beam dipole trap described in ref. [64]). To prepare the initial state of the BEC at the single minimum of the ground dressed band at $\Omega_I = 5.2\,E_r$ (at $\delta_R = \delta'(\Omega_I, \varepsilon)$, shown in Fig. 1c), first the Raman coupling $\Omega$ is ramped on slowly from 0 to $\Omega_I$ in 80 ms (slow enough compared to the trap period and any inter-band excitation process) with $\delta_R \sim -\varepsilon$ in Eq. (5), such that the dominant bare spin component of the dressed BEC at any finite $\Omega$ during the ramping process remains in $|m_F = 0\rangle$. Subsequently, while holding $\Omega$ at $\Omega_I$, we adjust $B$ to change the Raman detuning from $\delta_R \sim -\varepsilon$ to $\delta_R = \delta'(\Omega_I, \varepsilon)$ in 80 ms, and then we hold both $\Omega$ and $\delta_R$ for another 20 ms to let the system equilibrate. Note that adjusting $\delta_R$ to $\delta'(\Omega_I, \varepsilon)$ has to be empirically achieved by realizing the balanced spin populations, with the reasons addressed in the next paragraph. When the BEC is successfully prepared in the initial state at $\Omega_I$, equal populations in the $|m_F = -1, +\hbar k_r\rangle$ and $|m_F = 0, -\hbar k_r\rangle$ states can be achieved and seen in TOF images measured at $t_{hold} = -1$ ms.

In addition to the change in the band structure when going from the two-state picture to the three-state picture as discussed in the previous section, there are several other experimental factors that can lead to unbalanced spin populations. First, the slow drift in $\Omega$ can tilt (therefore unbalance) the band at a fixed $\delta_R$. Second, a slow drift in $B$ would give rise to a drift in $\delta_R$. Third, sometimes there may still be excitations (for example, small-amplitude collective dipole oscillations of a dressed BEC) at the end of the initial state preparation[65], making the quasimomentum of the dressed BEC deviate slightly from the quasimomentum of the band minimum. As a result, the dressed BEC can have a nonzero group velocity and unbalanced spin populations at $\Omega_I$ (before applying the spin-dependent electric fields $E_\sigma$). Hence, this can lead to unbalanced spin populations after the application of $E_\sigma$, and the spin polarization $P$ of atoms is not maintained around zero during $t_{hold}$. Here, we define $P = (N^\uparrow - N^\downarrow)/(N^\uparrow + N^\downarrow)$, where $N^{\uparrow(\downarrow)}$ is the total atom number of the atomic cloud (measured in the TOF images) for the bare spin component $\uparrow(\downarrow)$. Fourth, the quench process from the single minimum to double minima bands during $t_E$ (Fig. 1c) may also give rise to unbalanced spin populations, presumably because of the access to the magnetic phase in the double minima regime where the ground state is the occupation of a single dressed spin state (the two occupied dressed spin states are metastable states).

The above effects are avoided in our experiments by making sure that the balanced spin populations are empirically achieved throughout our experiment

(with occasional adjustment of $\delta_R$, and discarding runs with notably unbalanced spin populations). More specifically, we first make sure that balanced spin populations can be achieved at $\Omega_I$, assuring $\delta_R = \delta'(\Omega_I, \varepsilon)$ after the initial preparation described above. Then, we linearly ramp $\delta_R$ from $\delta'(\Omega_I, \varepsilon)$ to $\delta'(\Omega_F, \varepsilon)$ as we change $\Omega$ from $\Omega_I$ to $\Omega_F$ in $t_E$, and subsequently hold $\delta_R$ at $\delta'(\Omega_F, \varepsilon)$ for various $t_{hold}$. Here, $\delta_R = \delta'(\Omega_F, \varepsilon)$ is empirically achieved by realizing balanced spin populations at $\Omega = \Omega_F$ for various $t_{hold}$. Therefore, when we state $\delta_R = 0$ at a given $\Omega$ in the main text, it more precisely means that we realize balanced spin populations (as would be achieved at $\delta_R = 0$ in the 2-state picture described by Eq. (1)).

The above-mentioned procedure of realizing $\delta_R = \delta'(\Omega_F, \varepsilon)$ is further experimentally verified by observing balanced spin populations using the same bias magnetic fields but with $t_E = 15$ ms and $t_{hold} = 30$ ms (slow enough to not to excite notable SDM). This suggests that such a choice of $\delta_R = \delta'(\Omega_F, \varepsilon)$ approximates a balanced double minima band (with two equal-energy minima) at $\Omega_F$.

For the SDM measurements (e.g., Fig. 3), we make sure that the typical spin polarization is close to zero, with $|P| = 0.05 \pm 0.04$, where 0.05 is the mean and 0.04 is the standard deviation of the data. Note that we used the total atom numbers $N^{\uparrow(\downarrow)}$ instead of condensate atom numbers $N_c^{\uparrow(\downarrow)}$ to obtain $P$ due to the less fluctuation in the fitted $N^{\uparrow(\downarrow)}$. Typically images with such small $P$, indicating good spin population balance for the whole atomic cloud, also do not exhibit notable spin population imbalance in their condensate parts.

After holding the atoms in the trap at $\Omega_F$ for various $t_{hold}$, we turn off all lasers abruptly and do a 15-ms TOF, which includes a 9-ms Stern–Gerlach process in the beginning to separate the atoms of different bare spin states. Then, the absorption imaging is performed at the end of TOF to obtain the bare spin and momentum compositions of atoms. We then extract the physical quantities such as the mechanical momentum, condensate and thermal atom numbers of the atomic cloud in each spin state from such TOF images.

**Analysis of momentum damping.** Since the propagation direction ($\hat{x}'$) of our imaging laser is ~27° with respect to the x-axis in the $x - z$ plane (see Fig. 1a), the TOF images are in the $y - z'$ plane (where $\hat{z}'$ is perpendicular to $\hat{x}'$ in the $x - z$ plane). The atomic cloud of each (dominant) bare spin component in the TOF images is fitted to a 2D bimodal distribution:

$$A \max\left(1 - \left(\frac{y - y_c}{R_y}\right)^2 - \left(\frac{z' - z_c}{R_{z'}}\right)^2, 0\right)^{3/2}$$
$$+ B \exp\left(-\frac{1}{2}\left(\left(\frac{y - y_{cT}}{\sigma_y}\right)^2 + \left(\frac{z' - z_c}{\sigma_{z'}}\right)^2\right)\right) \quad (6)$$

where the first term corresponds to the condensate part according to the Thomas-Fermi approximation and the second term corresponds to the thermal part. Note that we only fit the majority bare spin cloud component when there is a distinguishable minority bare spin cloud component (which belongs to the same dressed spin state, but has a population <9% of the majority component in our experiments). This convention also applies to the analysis of the spin polarization defined above, condensate fraction, and the coherent spin current (see below). In the spin current or SOC directions ($\hat{y}$), we obtain the relative mechanical momentum between the two bare spin components $\hbar k_{spin} = \hbar(k_\uparrow - k_\downarrow)$ from the difference between the center-of-mass positions of their condensate parts ($y_c^\uparrow - y_c^\downarrow$) and the calibration of $2\hbar k_r$ in TOF images (for example, $2\hbar k_r$ can be calibrated from the distance between different bare spin components $\uparrow$ and $\downarrow$ that are in the same dressed spin state $\uparrow'$). To obtain the damping $(1/Q)$ of the relative momentum oscillations in SDM (Fig. 3), $\hbar k_{spin}$ as a function of $t_{hold}$ is fitted to a damped sinusoidal function $A_0 e^{-t_{hold}/\tau_{damp}} \cos(\omega t_{hold} + \theta_0) + B_0$, where $\tau_{damp}$ is the momentum decay time constant. The data have a small offset $B_0$ because we only use the majority bare spin component in each dressed spin state when extracting $\hbar k_{\uparrow,\downarrow}$. We extract $\tau_{damp}$ to obtain the inverse quality factor $1/Q = t_{trap}/(\pi \tau_{damp})$, where $t_{trap} = (2\pi/\omega_y)\sqrt{m_{eff}/m}$ is the trap period along the y direction taking into account of the effective mass $m_{eff}$ for the dressed band around $q_{\sigma \min}$, $m$ is the bare atomic mass, and $\omega_y/(2\pi)$ is the trap frequency along the y direction in the absence of Raman lasers. Note that the effective masses around the two minima in the dressed ground band are nearly the same so we take their average as the $m_{eff}$. The standard error of the fit (95% confidence intervals) is obtained for determining the uncertainty of $1/Q$ shown in Fig. 3 in the main text.

For the dipole oscillations of a BEC with a single dressed spin component prepared in the $|\downarrow'\rangle$ state (see Supplementary Note 1), we fit $\hbar k_\downarrow$ (mechanical momentum of the dominant bare spin component $|\downarrow\rangle$) as a function of $t_{hold}$ to a damped sinusoidal function to extract $\tau_{damp}$ and thus to obtain $1/Q$. The minority bare spin $|\uparrow\rangle$ component oscillates in phase with the dominant $|\downarrow\rangle$ component with similar damping, and thus is not taken into account for determining $1/Q$.

**Analysis of condensate fraction.** During the SDM, the atomic cloud can be significantly deformed along $\hat{y}$ due to the interference between the two dressed spin components (see e.g., Fig. 5). Therefore, in order to extract the total condensate fraction ($f_c = N_c/N$) of atoms to study the thermalization behavior as shown in Fig. 4a, b, the measured optical density (OD) of each bare spin component $\sigma$ in the $y - z'$ plane is integrated along the y direction (the direction of SOC and the spin

current as well as the direction along which the cloud can be significantly distorted) to obtain an integrated optical density versus $z'$ (denoted by $OD_{z'}$). We fit $OD_{z'}$ of each bare spin component $\sigma$ to a 1D bimodal distribution $A \max(1 - (\frac{z' - z_c}{R_{z'}})^2, 0)^2 + B \exp(-\frac{1}{2}(\frac{z' - z_c}{\sigma_{z'}})^2)$, where the first term corresponds to the condensate part according to the Thomas-Fermi approximation and the second term corresponds to the thermal part. By getting the corresponding condensate and thermal atom numbers, $N_c^\sigma$ and $N_{therm}^\sigma$, respectively. The total condensate fraction is calculated as $f_c = N_c/N = (N_c^\uparrow + N_c^\downarrow)/(N_c^\uparrow + N_{therm}^\uparrow + N_c^\downarrow + N_{therm}^\downarrow)$, shown as the scatters (unsmoothed raw data) in Fig. 4a.

To quantitatively describe the thermalization, we fit the smoothed total condensate fraction versus $t_{hold}$ to a shifted exponential decay $f_c(t) = f_s + (f_i - f_s) \exp(-t/\tau_{therm})$, where $\tau_{therm}$ represents the time constant for the thermalization to stop and for the decreasing condensate fraction to saturate, with $f_s$ being the saturation condensate fraction. Because the large fluctuations in the unsmoothed data can give erroneous fitting results, each fitted curve shown as a solid line in Fig. 4a is the average of the three fits performed on the smoothed data, obtained using different levels ($M = 1, 2, 3$) of smoothing, where the smoothing is done by taking the average of the raw data within the nearest $M$ time intervals.

Notice that the heating effect due to our Raman lasers (such as from spontaneous emission) is negligible within the time scale of the experiments (30 ms), because the lifetime of our BEC in the presence of the Raman lasers (with the Raman coupling considered in this work) is measured to be hundreds of ms. For example, the control experiment in Supplementary Fig. 1 shows no observable thermalization within 30 ms for dipole oscillations of a BEC with a single dressed spin component in the presence of the Raman lasers.

**Coherent spin current.** The $I_\sigma$ in Eq. (2) reflects the number of BEC atoms of a specific spin state passing through a cross section per unit time, and can be related to $JA$, where $J = n_c v$ is the current density along the SOC direction ($\hat{y}$) with the effective number density $n_c = N_c/(LA)$, $v$ is the corresponding velocity, and $A$ is an effective cross sectional area (the spin index $\sigma$ is dropped in this discussion for simplicity in notations). The in situ length in the $y$ direction, $L$, of each bare spin component can be estimated from the measured length of the BEC after TOF by $L_y(t_{TOF}) = \sqrt{1 + (\omega_y t_{TOF})^2} L_y(t_{TOF} = 0)$ for a cigar-shape interacting BEC with $\omega_{x,y} \gg \omega_z$ and in the Thomas-Fermi approximation[66], where $L_y(t_{TOF})$ is defined as $2R_y$ in Eq. (6) and $L_y(t_{TOF} = 0) = L$. For example, for a typical $L_y(t_{TOF} = 15$ ms$) = 88$ $\mu$m measured for one bare spin component of a dressed BEC prepared at $\Omega_I$, we get $L = 4.5$ $\mu$m for $\omega_y = 2\pi \times 205$ Hz. The two spin components have similar $L$ when the spin populations are balanced. The in situ length $L$ is $t_{hold}$-dependent during the dynamics and calculated from the $t_{hold}$-dependent TOF size, and is then used to obtain the $t_{hold}$-dependent spin current in Fig. 4b.

In the Thomas-Fermi approximation, we can also calculate $L$ for the initial state at $\Omega_I$ from the condensate atom number and trap frequencies. For example, we obtain $L = 4.7$ $\mu$m using $N_c = 1.6 \times 10^4$ and $\omega_y = 2\pi \times 205$ Hz by $\mu = \frac{1}{2}m\omega_y^2 L^2$, where $\mu = \frac{15^{\frac{2}{5}}}{2}(N_c a/\bar{a})^{\frac{2}{5}}\hbar\bar\omega$, $\bar\omega = (\omega_x \omega_y \omega_z)^{1/3}$, $a$ is the s-wave scattering length, and $\bar{a} = \sqrt{\hbar/(m\bar\omega)}$. In addition, the GPE-simulated $L$ is 4.7 $\mu$m. These results of $L$ are consistent with the value calculated from the TOF width.

**Analysis of BEC shape oscillations.** We characterize a condensate's shape oscillations (in the $y - z'$ plane) of the bare spin component $\sigma$ by its aspect ratio $W_y^\sigma/W_{z'}^\sigma$, where the condensate widths $W_y^\sigma = 2R_y^\sigma$ and $W_{z'}^\sigma = 2R_{z'}^\sigma$ (respectively along the $y$ and $z'$ directions) are obtained from Eq. (6). We take the average of the aspect ratios of the two spin components $(W_y/W_{z'} = (W_y^\uparrow/W_{z'}^\uparrow + W_y^\downarrow/W_{z'}^\downarrow)/2)$, and plot $W_y/W_{z'}$ as a function of $t_{hold}$ in Fig. 5. In Fig. 5e, caution has to be paid because the prominent thermalization in the bare case can make it challenging to fit the 2D cloud and extract the aspect ratio. The notable distortion of the cloud at the early stages of SDM can also make it difficult to perform the 2D Thomas–Fermi fit. Therefore, in Fig. 5f, we choose the $t_{hold}$-dependent $W_y/W_{z'}$ data after the corresponding dashed line (indicating $t_{hold} \sim 2\tau_{damp}$ after which the SDM is fully damped) to fit to a damped sinusoidal function to extract the frequency of the aspect ratio oscillations.

In our experiments, there is no external modulation of the trapping potentials or shapes of the BECs to intentionally excite the shape oscillations. However, it is worth noting that shape oscillations can be induced via a non-adiabatic change in the internal energy of atomic clouds[67,68], which can take place when $\Omega$ is quickly changed or when the two spin components collide within the trap. On the other hand, we notice that in the dressed case the formation of density modulations can significantly deform the shape of a BEC (Fig. 5b, d; Supplementary Movies 2, 4 and 5 in Supplementary Note 3) and may thus also induce energetically allowed BEC shape oscillations, because the modified shape of the atomic cloud is no longer in equilibrium with the trap. Note that such a shape deformation can also change the internal energy. The $m = 0$ quadrupole mode excitation observed in our experiments has the lowest mode frequency among all possible quadrupole modes given our trap geometry and thus is the most energetically favorable (its mode frequency is also lower than the SDM frequency $\sim\omega_y/(2\pi)$ for our trap parameters). Such nonresonant mode excitation is quite different from most previous studies, in which a collective mode of an atomic cloud is efficiently excited when it matches with the external modulation or perturbation of the trap[53,69] spatially and also

spectrally (resonant with the modulation frequency). Compared to the dressed case, the bare case has less damped SDM and more significant thermalization, thus may complicate the shape oscillations due to more repeated SDM collisions and more atom loss[67,68]. We expect that the energy of the shape oscillations may eventually be converted to the energy of thermal atoms, leading to decay of the collective modes.

To further verify the excitation of the $m = 0$ quadrupole mode in the dressed case, we used another set of trap frequencies (see Supplementary Fig. 3 in Supplementary Note 2), and measured the condensate's aspect ratio as a function of $t_{hold}$. The extracted frequency for the aspect ratio oscillations is again consistent with the predicted frequency for the $m = 0$ quadrupole mode.

**Calculation of nonorthogonality, effective interaction parameters, and immiscibility.** The interactions between atoms in bare spinor BECs are characterized by the interspecies ($g_{\uparrow\downarrow}$, $g_{\downarrow\uparrow}$) and intraspecies ($g_{\uparrow\uparrow}$, $g_{\downarrow\downarrow}$) interaction parameters, where $g_{\downarrow\downarrow} = g_{\uparrow\downarrow} = g_{\downarrow\uparrow} = \frac{4\pi\hbar^2(c_0+c_2)}{m}$, $g_{\uparrow\uparrow} = \frac{4\pi\hbar^2 c_0}{m}$, $c_2 = -0.46a_0$, and $c_0 = 100.86a_0$ ($a_0$ is the Bohr radius) for $^{87}$Rb atoms in our case. For a dressed BEC, in which $|\uparrow '\rangle$ is at some quasimomentum $\hbar q_y$ (>0) and $|\downarrow '\rangle$ is at $-\hbar q_y$ in the ground dressed band at $\Omega$ (in the two-state picture described by Eq. (1) in the main text with $\delta_R = 0$), the effective interspecies ($g_{\uparrow'\downarrow'} = g_{\downarrow'\uparrow'}$) and intraspecies ($g_{\uparrow'\uparrow'}$, $g_{\downarrow'\downarrow'}$) interaction parameters can be expressed in terms of the bare interaction g-parameters:

$$g_{\uparrow'\uparrow'} = \frac{g_{\uparrow\uparrow}}{4}\left(1+\cos\theta_{q_y}\right)^2 + \frac{g_{\downarrow\downarrow}}{4}\left(1-\cos\theta_{q_y}\right)^2 \\ + \frac{g_{\uparrow\downarrow}}{2}\left(1-\cos^2\theta_{q_y}\right) \tag{7}$$

$$g_{\downarrow'\downarrow'} = \frac{g_{\uparrow\uparrow}}{4}\left(1-\cos\theta_{q_y}\right)^2 + \frac{g_{\downarrow\downarrow}}{4}\left(1+\cos\theta_{q_y}\right)^2 \\ + \frac{g_{\uparrow\downarrow}}{2}\left(1-\cos^2\theta_{q_y}\right) \tag{8}$$

$$g_{\uparrow'\downarrow'} = \frac{g_{\uparrow\uparrow}+g_{\downarrow\downarrow}}{2}\left(1-\cos^2\theta_{q_y}\right) + g_{\uparrow\downarrow} \tag{9}$$

where $\cos\theta_{q_y} = \left(\hbar^2 q_y k_r/m\right)/\sqrt{\hbar^4 q_y^2 k_r^2/m^2 + (\Omega/2)^2}$. The dressed spin states $|\downarrow '\rangle$ at $-\hbar q_y$ and $|\uparrow '\rangle$ at $\hbar q_y$ in the ground dressed band can be expressed as

$$|\downarrow '\rangle = \begin{pmatrix} \cos\left(\frac{\theta_{q_y}}{2}\right) \\ -\sin\left(\frac{\theta_{q_y}}{2}\right) \end{pmatrix} \tag{10}$$

$$|\uparrow '\rangle = \begin{pmatrix} \sin\left(\frac{\theta_{q_y}}{2}\right) \\ -\cos\left(\frac{\theta_{q_y}}{2}\right) \end{pmatrix} \tag{11}$$

in the bare spin basis of $\{|\downarrow\rangle, |\uparrow\rangle\}$. Using Eqs. (10) and (11), we can further obtain

$$\langle\uparrow '|\downarrow '\rangle = \sin\theta_{q_y} = (\Omega/2)/\sqrt{\hbar^4 q_y^2 k_r^2/m^2 + (\Omega/2)^2} \tag{12}$$

which characterizes the nonorthogonality (and thus the interference) between the two dressed spin states (where $|\uparrow '\rangle$ is located at $\hbar q_y$ and $|\downarrow '\rangle$ is located at $-\hbar q_y$ in the ground dressed band at $\Omega$). Figure 7a plots such nonorthogonality versus quasimomentum for various $\Omega$.

Note that $\theta_{q_y}$ (which is between 0 and $\pi/2$ in our case) characterizes the degree of bare spin mixing of a single dressed spin state (Eqs. (10) and (11)) as well as the nonorthogonality (due to the bare spin mixing, see Eq. (12)) between the two dressed spin states. As we can see, either decreasing $\Omega$ or increasing $q_y$ would decrease $\theta_{q_y}$ (or increase $\cos\theta_{q_y}$). When $\theta_{q_y} \to 0$ (or $\cos\theta_{q_y} \to 1$), all the dressed spin states would approach the corresponding bare spin states, i.e., $|\uparrow '\rangle \to |\uparrow\rangle$ and $|\downarrow '\rangle \to |\downarrow\rangle$, thus the nonorthogonality $\langle\uparrow'|\downarrow'\rangle \to 0$. In addition, all the effective interaction parameters would approach the corresponding bare values, i.e., $g_{\uparrow'\uparrow'} \to g_{\uparrow\uparrow}$, $g_{\downarrow'\downarrow'} \to g_{\downarrow\downarrow}$, and $g_{\uparrow'\downarrow'} \to g_{\uparrow\downarrow}$.

On the other hand, either increasing $\Omega$ or decreasing $q_y$ would increase $\theta_{q_y}$ towards $\pi/2$ (or decrease $\cos\theta_{q_y}$), thus enhancing the bare spin mixing, nonorthogonality and $g_{\uparrow'\downarrow'}$. When $\theta_{q_y} \to \pi/2$ (or $\cos\theta_{q_y} \to 0$),

$g_{\uparrow'\uparrow'} \to \frac{g_{\uparrow\uparrow}}{4} + \frac{g_{\downarrow\downarrow}}{4} + \frac{g_{\uparrow\downarrow}}{2}$, $g_{\downarrow'\downarrow'} \to \frac{g_{\uparrow\uparrow}}{4} + \frac{g_{\downarrow\downarrow}}{4} + \frac{g_{\uparrow\downarrow}}{2}$, and $g_{\uparrow'\downarrow'} \to \frac{g_{\uparrow\uparrow}}{2} + \frac{g_{\downarrow\downarrow}}{2} + g_{\uparrow\downarrow}$. Therefore, $g_{\uparrow'\downarrow'} \to 2g_{\uparrow'\uparrow'}$ or $2g_{\downarrow'\downarrow'}$, which is the upper bound of the effective interspecies interaction parameter. Figure 7b, c shows the effective interaction parameters normalized by $g_{\uparrow\uparrow}$ versus quasimomentum $\hbar q_y$ at $\Omega = 0.1\,E_r$ and $\Omega = 1.26\,E_r$, respectively. When $\Omega$ increases or $q_y$ decreases, $g_{\uparrow'\downarrow'}$ increases while $g_{\uparrow'\uparrow'}$ and $g_{\downarrow'\downarrow'}$ almost remain at the bare values. As $q_y \to 0$ at any finite $\Omega$, $g_{\uparrow'\downarrow'}$ approaches the upper limit $2g_{\uparrow'\uparrow'}$ or $2g_{\downarrow'\downarrow'}$.

In the case of SDM, assume that in the ground dressed band at $\Omega$, $|\uparrow '\rangle$ is located at $\hbar q_y$ and $|\downarrow '\rangle$ is located at $-\hbar q_y$ at $t_{hold}$, we may use the immiscibility

metric[70]

$$\eta = \left(g_{\uparrow'\downarrow'}^2 - g_{\uparrow'\uparrow'}g_{\downarrow'\downarrow'}\right)/g_{\uparrow\uparrow}^2 \tag{13}$$

to understand how $\Omega$ may modify the miscibility ($\eta < 0$) or immiscibility ($\eta > 0$) between $|\uparrow '\rangle$ and $|\downarrow '\rangle$.

**GPE simulations.** The dynamical evolution of a BEC is simulated by the 3D time-dependent GPE[71]. To compare with the experimental data, we conduct simulations with similar parameters as those used in our experiment. The GPE of a SO-coupled BEC can be written in the following form:

$$i\hbar\frac{\partial}{\partial t}\Psi(\mathbf{r},t) = H_{tot}\Psi(\mathbf{r},t) \\ = \left(\frac{\hat{p}_x^2}{2m} + \frac{\hat{p}_z^2}{2m} + H_{SOC} + V_{trap} + V_{int}\right)\Psi(\mathbf{r},t) \tag{14}$$

where $\hat{p}_x = -i\hbar\frac{\partial}{\partial x}$ ($\hat{p}_z = -i\hbar\frac{\partial}{\partial z}$) is the momentum operator along $\hat{x}(\hat{z})$, and $H_{SOC}$ is the (two-state) single-particle Hamiltonian Eq. (1), with $q_y$ replaced by $\hat{q}_y = \hat{p}_y/\hbar = -i\frac{\partial}{\partial y}$. $V_{trap}$ is the external trapping potential:

$$V_{trap} = \frac{1}{2}m\omega_x^2 x^2 + \frac{1}{2}m\omega_y^2 y^2 + \frac{1}{2}m\omega_z^2 z^2 \tag{15}$$

where $\omega_{x(y,z)}$ is the angular trap frequency along the spatial coordinate $x(y, z)$. The wavefunction (order parameter) of a spinor BEC can be written in the form

$$\Psi = \begin{pmatrix} \psi_\downarrow \\ \psi_\uparrow \end{pmatrix} = \begin{pmatrix} \sqrt{n_\downarrow(\mathbf{r},t)}e^{i\phi_\downarrow(\mathbf{r},t)} \\ \sqrt{n_\uparrow(\mathbf{r},t)}e^{i\phi_\uparrow(\mathbf{r},t)} \end{pmatrix} \tag{16}$$

where $\psi_\downarrow$ and $\psi_\uparrow$ are the respective condensate wavefunctions of the two components, $n_\downarrow(n_\uparrow)$ is the condensate density, $\phi_\downarrow(\phi_\uparrow)$ is the phase of the wavefunction, $\mathbf{r}$ is the position, and $t$ is time. The spatial integration of $(n_\downarrow + n_\uparrow)$ gives the total atom number $N$. The two-body interactions between atoms are described by the nonlinear interaction term $V_{int}$, which can be written in the basis of $\{\psi_\downarrow, \psi_\uparrow\}$:

$$V_{int} = \begin{pmatrix} g_{\downarrow\downarrow}\left|\psi_\downarrow\right|^2 + g_{\uparrow\downarrow}\left|\psi_\uparrow\right|^2 & 0 \\ 0 & g_{\uparrow\uparrow}\left|\psi_\uparrow\right|^2 + g_{\uparrow\downarrow}\left|\psi_\downarrow\right|^2 \end{pmatrix} \tag{17}$$

The interaction parameters are given by

$$g_{\downarrow\downarrow} = g_{\uparrow\downarrow} = g_{\uparrow\downarrow} = \frac{4\pi\hbar^2(c_0+c_2)}{m} \tag{18}$$

and

$$g_{\uparrow\uparrow} = \frac{4\pi\hbar^2 c_0}{m} \tag{19}$$

The spin-dependent s-wave scattering lengths for $^{87}$Rb atoms are $c_0$ and $c_0 + c_2$, where $c_2 = -0.46a_0$ and $c_0 = 100.86a_0$ ($a_0$ is the Bohr radius). The initial state of the SO-coupled BEC is obtained by using the imaginary time propagation method. Next we change $\Omega_I$ to a final value $\Omega_F$ at $t_E = 1.0$ ms to simulate the spin-dependent synthetic electric fields. Equation (14) is used to simulate the dynamics of the BECs. The momentum space wavefunctions are calculated from the Fourier transformation of the real space wave functions. The squared amplitude of the momentum space wavefunctions is used to obtain the time-dependent momentum space density distributions shown in e.g., Fig. 6a, b.

For the GPE simulations in Fig. 6, we have checked that moderate variations in these parameters (as in our experimental data) do not affect our conclusions (while they can slightly change the $1/Q$ values, for example, larger $1/Q$ found for higher $N_c$). The simulations also reveal additional interesting features, such as the appearance of the opposite momentum (back-scattering) peak for each spin component in Fig. 6a, b, which are not well resolved in our experimental data.

**Different forms of energies in GPE simulations.** Using Eq. (16), the total energy density $\varepsilon$ (the spatial integration of which gives the total energy of the system) can

be expressed as the sum of several terms[35,59]:

$$\varepsilon = \varepsilon_1 + \varepsilon_2 + \varepsilon_3 + \varepsilon_4 + \varepsilon_5 \tag{20}$$

$$\varepsilon_1 = \frac{\hbar^2}{8mn_\downarrow}\left(\nabla n_\downarrow\right)^2 + \frac{\hbar^2}{8mn_\uparrow}\left(\nabla n_\uparrow\right)^2 \tag{21}$$

$$\varepsilon_2 = \frac{\hbar^2 n_\downarrow}{2m}\left(\nabla\phi_\downarrow\right)^2 + \frac{\hbar^2 n_\uparrow}{2m}\left(\nabla\phi_\uparrow\right)^2 \\ + \frac{\hbar^2 k_r}{m}\left(n_\downarrow\nabla_y\phi_\downarrow - n_\uparrow\nabla_y\phi_\uparrow\right) + \frac{\hbar^2 k_r^2}{2m}\left(n_\downarrow + n_\uparrow\right) \tag{22}$$

$$\varepsilon_3 = \Omega\sqrt{n_\downarrow n_\uparrow}\cos\left(\phi_\downarrow - \phi_\uparrow\right) \tag{23}$$

$$\varepsilon_4 = \frac{g_{\downarrow\downarrow}}{2}\left(n_\downarrow\right)^2 + \frac{g_{\uparrow\uparrow}}{2}\left(n_\uparrow\right)^2 + g_{\downarrow\uparrow}n_\downarrow n_\uparrow \tag{24}$$

$$\varepsilon_5 = V_{trap}\left(n_\downarrow + n_\uparrow\right) \tag{25}$$

In the above equations, $\nabla = \frac{\partial}{\partial x}\hat{x} + \frac{\partial}{\partial y}\hat{y} + \frac{\partial}{\partial z}\hat{z}$ and $\nabla_y = \frac{\partial}{\partial y}$. We will introduce $\varepsilon_1$ to $\varepsilon_5$ one by one in the following. The expression of $\varepsilon_1$ in Eq. (21) is the density of the total (including two spin components) quantum pressure (QP), which is a type of kinetic energy (KE) associated with the spatial variation of the condensate density. An imaginary term $-\frac{i\hbar^2 k_r}{m}\nabla_y(n_\downarrow - n_\uparrow)$ appearing in the derivation of $\varepsilon_1$ is not shown in Eq. (21) as its spatial integration (for a confined system) is zero and thus has no contribution to the energy. The expression of $\varepsilon_2$ in Eq. (22) is the density of the sum of two types of KE, the total CoM KE (sum of the CoM KE of both bare spin components) and the total local current kinetic energy (LC KE). Both the CoM KE and LC KE are associated with the spatial variation of the phase of wavefunctions. The sum of the three types of kinetic energy (total QP, total CoM KE, and total LC KE) gives the total KE. That is, the sum of $\varepsilon_1$ and $\varepsilon_2$ is the density of the total KE. In the following, we derive explicit expressions for the CoM KE and LC KE. For CoM KE, it is nonzero only in the $y$ direction because the SDM is along the $y$ direction. Thus, the expression of CoM KE is:

$$\text{CoM KE} = \frac{1}{2m}\left(\left\langle\psi_\downarrow\left|\hbar\hat{k}_\downarrow\right|\psi_\downarrow\right\rangle^2 + \left\langle\psi_\uparrow\left|\hbar\hat{k}_\uparrow\right|\psi_\uparrow\right\rangle^2\right) \tag{26}$$

$$= \frac{\hbar^2}{2m}\left(\left\langle\psi_\downarrow\left|\nabla_y\phi_\downarrow\right|\psi_\downarrow\right\rangle^2 + \left\langle\psi_\uparrow\left|\nabla_y\phi_\uparrow\right|\psi_\uparrow\right\rangle^2\right) \\ + \frac{\hbar^2 k_r}{m}\left(\left\langle\psi_\downarrow\left|\nabla_y\phi_\downarrow\right|\psi_\downarrow\right\rangle - \left\langle\psi_\uparrow\left|\nabla_y\phi_\uparrow\right|\psi_\uparrow\right\rangle\right) \\ + \left\langle\Psi\left|\frac{\hbar^2 k_r^2}{2m}\right|\Psi\right\rangle, \tag{27}$$

where $\hbar\hat{k}_\downarrow = \hbar(\hat{q}_y + k_r) = \hbar(-i\frac{\partial}{\partial y} + k_r)$ $(\hbar\hat{k}_\uparrow = \hbar(\hat{q}_y - k_r) = \hbar(-i\frac{\partial}{\partial y} - k_r))$ is the momentum operator along $\hat{y}$ for the spin down (up) component, and the last term in Eq. (27) is simply $N\frac{\hbar^2 k_r^2}{2m}$. Recall that $\varepsilon_2$ in Eq. (22) is the density of the sum of CoM KE and LC KE. Thus, the expression of LC KE can be obtained by subtracting the expression of CoM KE in Eq. (27) from the spatial integration of $\varepsilon_2$ (Eq. (22)):

$$\text{LC KE} = \frac{\hbar^2}{2m}\left(\left\langle\psi_\downarrow\left|\left(\nabla\phi_\downarrow\right)^2\right|\psi_\downarrow\right\rangle + \left\langle\psi_\uparrow\left|\left(\nabla\phi_\uparrow\right)^2\right|\psi_\uparrow\right\rangle\right) \\ - \frac{\hbar^2}{2m}\left(\left\langle\psi_\downarrow\left|\nabla_y\phi_\downarrow\right|\psi_\downarrow\right\rangle^2 + \left\langle\psi_\uparrow\left|\nabla_y\phi_\uparrow\right|\psi_\uparrow\right\rangle^2\right) \tag{28}$$

$$= \frac{\hbar^2}{2m}\left(\Delta\left(\nabla_x\phi_\downarrow\right) + \Delta\left(\nabla_x\phi_\uparrow\right) + \Delta\left(\nabla_z\phi_\downarrow\right) \\ + \Delta\left(\nabla_z\phi_\uparrow\right) + \Delta\left(\nabla_y\phi_\downarrow\right) + \Delta\left(\nabla_y\phi_\uparrow\right)\right), \tag{29}$$

where $\Delta(\nabla_{x,y,z}\phi_{\downarrow,\uparrow})$ is the standard deviation of $\nabla_{x,y,z}\phi_{\downarrow,\uparrow}$, and note $\langle\nabla_{x,z}\phi_{\downarrow,\uparrow}\rangle = 0$. Thus, if the wavefunction is a plane wave with a phase $\phi = q_y y$, its LC KE is zero. For collective modes that do not have the CoM KE (for example, the quadrupole modes), the associated motional (kinetic) energy can be accounted for by LC KE and QP. The expression of $\varepsilon_3$ in Eq. (23) is the density of the Raman energy, associated with the Raman coupling $\Omega$. The expression of $\varepsilon_4$ in Eq. (24) is the density of the sum of the bare intraspecies and interspecies interaction energies. The expression of $\varepsilon_5$ in Eq. (25) is the density of the total potential energy.

To calculate the time ($t_{hold}$) evolution of the various forms of energies, we can in principle integrate the corresponding time-dependent energy densities over the real space. In practice, for the kinetic energy part we only perform spatial integration of $\varepsilon_2$ (given by Eq. (22)). For convenience of computation, the total KE, total CoM KE, total LC KE, and total QP are calculated using a different approach taking

advantages of the (quasi)momentum space representation of the quantum mechanical wavefunctions and operators. Specifically, the total KE is calculated by $\langle\psi_\downarrow(\mathbf{q},t)|\frac{(\hbar\hat{k}_\downarrow)^2+\hat{p}_x^2+\hat{p}_z^2}{2m}|\psi_\downarrow(\mathbf{q},t)\rangle + \langle\psi_\uparrow(\mathbf{q},t)|\frac{(\hbar\hat{k}_\uparrow)^2+\hat{p}_x^2+\hat{p}_z^2}{2m}|\psi_\uparrow(\mathbf{q},t)\rangle$ in the quasimomentum space, where $\hbar\hat{k}_\downarrow = \hbar(\hat{q}_y + k_r)$ $(\hbar\hat{k}_\uparrow = \hbar(\hat{q}_y - k_r))$ is the momentum operator along $\hat{y}$ for the spin down (up) component, and $\psi_{\downarrow,\uparrow}(\mathbf{q},t)$ is the momentum-space representation of the wavefunctions (in the two directions not affected by SOC, $x$ and $z$, we simply have $q_x = p_x$ and $q_z = p_z$). Similarly, the total CoM KE is calculated in the quasimomentum space using Eq. (26). The total LC KE is calculated by subtracting the calculated total CoM KE from the spatial integration of $\varepsilon_2$ (Eq. (22)). The total QP is calculated indirectly by subtracting the spatial integration of $\varepsilon_2$ from the total KE.

The total Raman energy is calculated by the spatial integration of $\varepsilon_3$ (Eq. (23)). The total bare intraspecies ($g_{\uparrow\uparrow}$ and $g_{\downarrow\downarrow}$) and interspecies ($g_{\uparrow\downarrow}$) interaction energies are calculated by the spatial integration of the corresponding terms in $\varepsilon_4$ (Eq. (24)). The total interaction energy is calculated as the sum of the bare intraspecies and interspecies interaction energies. The total potential energy is calculated by the spatial integration of $\varepsilon_5$ (Eq. (25)). Lastly, the total energy of the system is calculated as the sum of the total Raman energy, total potential energy, total interaction energy, and total KE.

We note that even though our GPE simulations do not treat thermalization and thermal energies, the calculated different forms of condensate energies and their time evolution still provide valuable insights to understand the dynamical processes involved in the SDM. The GPE calculated different forms of energies shown in Fig. 8 and discussed in the associated texts below refer to the energies per particle (i.e., the calculated energies divided by the total atom number $N$).

In Fig. 8a, the total energy is a constant during $t_{hold}$, confirming the conservation of the total energy. In Fig. 8b, the total Raman energy has relatively small variations during $t_{hold}$. In Fig. 8c, the total potential energy in dressed cases has smaller variations during $t_{hold}$ compared with that in the bare case. In Fig. 8d, the time evolution of the total interaction energy at different $\Omega_F$ possesses a complicated behavior, mainly due to the complicated dynamics of the densities of the two spin components as well as their spatial overlap (see Supplementary Movies 2, 4 and 5 in Supplementary Note 3).

Figure 8e–h shows the time evolution of the calculated total KE, total CoM KE, total QP, and total LC KE at different $\Omega_F$, respectively. When $\Omega_F$ is larger, the total CoM KE (Fig. 8f) exhibits a faster damping while QP as well as LC KE exhibit a faster increase (Fig. 8g, h, focusing on the relatively early stage of SDM) presumably due to the enhancement of the interference, immiscibility, and effective interaction between the two dressed spin components.

Figure 8i–k shows the time evolution of the calculated intraspecies and interspecies interaction energies at different $\Omega_F$. Note that the interaction energies are relatively small compared to other forms of energies, but are essential for the damping mechanisms as discussed in the main text.

## Data availability

The data presented in this work are available from the corresponding author upon reasonable request.

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

## Acknowledgements

We thank Hui Zhai for helpful discussions and Ting-Wei Hsu for his help in experiments. This work has been supported in part by the Purdue University OVPR Research Incentive Grant and the NSF grant PHY-1708134. D.B.B. also acknowledges support by the Purdue Research Foundation Ph.D. fellowship. C.Q. and C.Z. are supported by NSF (PHY-1505496, PHY-1806227), ARO (W911NF-17-1-0128), and AFOSR (FA9550-16-1-0387). M.H. and Q.Z. acknowledge support from Hong Kong Research Council through CRF C6026-16W and start up funds from Purdue University. S.J.W. and C.H.G. are supported by NSF grant PHY-1607180. Y.L.-G. was supported by the U.S. Department of Energy, Office of Basic Energy Sciences, under Award DE-SC0010544.

## Author contributions

C.H.L., R.J.N., D.B.B., A.O., and Y.P.C. contributed to the experiment. C.Q. and C.Z. contributed to the GPE simulations. M.H. and Q.Z. contributed to the computations for the effective interactions. S.J.W., C.H.G., and Y.L.-G. contributed to additional theoretical insights. Y.P.C. supervised the work. All authors contributed to the physical interpretation for the results and to the writing of the manuscript.

## Additional information

**Competing interests:** The authors declare no competing interests.

