## [Peer Review File · Nature Communications]

Reviewers' comments:

Reviewer #1 (Remarks to the Author):

In the present paper the authors are presenting experimental results and theoretical analysis on the behaviour of the spin-dipole mode in a two-component Bose-Einstein condensate. In particular, after a quench in the Rabi-coupling of the spin-orbit lasers they observe the relaxation of the spin-dipole mode and the thermalization of the condensate fraction as a function of the final Rabi coupling after the quench. The experimental results are also compared with numerical simulations using the two-component Gross-Pitaevskii equation. The spin-dipole mode has been observed before, but what is new in this paper is the study of the effect of the spin-orbit coupling.

The paper is reasonable well written (but see below) and will be mostly interesting to the community working on ultracold gases. The authors claim that their results are of much broader interest, but that is not so obvious because in spintronics one is most often interested in fermionic (electronic) systems and here we are dealing with bosons. That community is presently also interested in magnon spintronics, which indeed deals with bosons, but the description is then based on the Landau-Gilbert equation, which is quite different from the Gross-Pitaevskii equation which is relevant for atomic Bose-Einstein condensates. It should be noted here that the Gross-Pitaevskii equation also does not include damping and thermalization, and so it is not expected that this description will be sufficient to understand all the details of this experiment. So presumably specific details of the atomic gas, such as the thermal cloud, will be important and this further reduces the overlap with the magnon systems.

Having said that I believe this paper presents new and interesting results that deserve to be published. It is timely in the sense that it deals both with spin-orbit coupling and nonequilibrium transport in an atomic Bose gas. My main objection with this paper is its present form. I believe that the paper would be much more readable and would have much more impact if the authors would write a single text that combines both the main text and the supplementary material. For instance, it is very important for this paper that the experimental results are compared with experiment, especially because the theory is expected not to be sufficiently accurate for all the experimental details. (For instance, there is no thermal cloud in the Gross-Pitaevskii equation, which will be crucial for damping and thermalization.) However, this comparison is only made in the supplementary material and not in the main text. Also the details of the fits of the shape of the clouds of the two spin components are crucial for understanding the final results on the thermalization and these are again only presented in the supplementary material. In my opinion, a combined text would be much more suited for publication and would be much helpful for the ultracold gases community, because the main text by itself does not give the necessary information to fully appreciate the results obtained by the authors.

Reviewer #2 (Remarks to the Author):

Summary of the key results:

In the manuscript "Spin Current Generation and Relaxation in a Quenched Spin-Orbit Coupled Bose-Einstein Condensate", the authors report on effects of synthetic SOC and spin transport using two interacting BECs of different spin states. These results are certainly worthy of publication in Nature Communications (with revisions). Mainly due to the substantive claim that SOC can significantly enhance SDM damping and reduce thermalization of BEC.

claims made:

- 1) Out-of-phase oscillations (spin-dipole mode, SDM) of the two BECs account for the spin current. — trivial claim
- 2) a) SOC can significantly enhance SDM damping and b) reduce thermalization of BEC — substantive claim
- 3) observe generation of diff BEC collective excitations, such as shape oscillations. — trivial claim
- 4) work may advance fundamental understanding of spin transport and quenched dynamics in SOCed systems.

The system consists of Rb-87 BECs in two hyperfine ground states that are then Raman coupled (Fig1b) to create synthetic 1D SOC with $|m_F=-1\rangle$ and $|m_F=0\rangle$ spin states. The BEC is initially prepared around the single minimum of the ground dressed band and after a hold time the final Raman coupling is quickly lowered to a final value that puts the BEC into the double minimum regime.

SDM oscillations are observed as a function of hold time at various final Raman couplings and show that at higher final Raman coupling, these oscillations are damped significantly which supports claim 2a. (Fig3 a-f). They further this claim and 2b in Fig4a-c by showing that less heating leads to an increase in condensate fraction.

To support claims 1), 2a-b), they complete control experiments:

Control exp1 - dipole oscillations of a SOCed BEC w single dressed spin comp: very small damping and negligible thermalization. — Spin current and net mass current

Control exp2 - exciting in-phase dipole oscillations of 2 dressed spin components of a SOCed BEC w/out relative collisions —> very small damping and negligible thermalization.— AC mass current with spin current.

In regards to claim 3) thermalization and spin current, It is not apparent in the text why the basic physics is happening. Adding a few lines to explain/describe such physics would benefit the non-expert reader.

claim 4) may contribute to vs advance... advance is a strong word here. Contribute to the understanding of quenched dynamics in SOCed systems, yes. Of course this is speculative, however, the final line of the conclusion, "may offer rich opportunities to study non-equilibrium quantum dynamics and novel superfluids in SOCed system"... is very vague.

Originality and significance: if not novel, please include reference:

Building off of prior methods, mainly refs [16,22], authors take it a step further and study spin current with two interacting BECs of different spins states. To my knowledge, this is the first study of in phase and out of phase oscillations of two interacting BECs demonstrating that Out-of-phase oscillations (spin-dipole mode, SDM) of the two BECs account for the spin current. This work has the added and substantive novelty that they understand a new mechanism for damping that is not of immediate intuitive transparency.

Data & methodology: validity of approach, quality of data, quality of presentation

The authors explain their methodology very clearly and openly discuss the caveats of their approach.

Appropriate use of statistics and treatment of uncertainties

Error bars on Raman coupling, condensate fraction and thermalization time seem reasonable, making the treatment of uncertainties clear.

Conclusions: robustness, validity, reliability

The data and supplemental documentation support the conclusions the authors have made.

Suggested improvements: experiments, data for possible revision

No additional experiments or data is needed to support the claims.

References: appropriate credit to previous work?

Yes. Perhaps the following paper is also a relevant reference: 2011a_Generation of Dark-Bright Soliton Trains in Superfluid-Superfluid.pdf
<https://arxiv.org/abs/1005.2610>

Reviewer #3 (Remarks to the Author):

In this paper, effects of spin-orbit (SO) coupling on a spin current in a Bose-Einstein condensate (BEC) are investigated. By rapidly changing the strength of engineered SO coupling, the authors drive a spin current consisting of out-of-phase oscillations of two BECs of different spin states, and they experimentally observe the subsequent dynamics with and without SO coupling. From these measurements they conclude that the effects of SO coupling are (i) enhancement of damping of the spin-dipole oscillations and (ii) reduction of thermalization. The first effect (i) is confirmed by numerical simulations based on the time-dependent Gross-Pitaevskii equation. Furthermore, by investigating the time evolution of different contributions of energy as well as the spatial distribution of the BEC wavefunctions in their simulation, the authors reveal that the spatial modulation of the amplitude and phase of the wavefunctions is the reason for the strong damping of the spin-dipole oscillations.

The second effect (ii) sounds more interesting, because the strong damping is usually considered to contribute to the thermalization. But here I am skeptical about the authors' conclusion. What they experimentally observed is that the BEC fraction becomes smaller for the dynamics without SO coupling. I simply considered that the change in SO coupling strength to trigger the dynamics is larger in the case of dynamics without SO coupling, leading to more energy input (and then more heating) to the system. The reduction of the BEC fraction somehow "normalized" by the energy input via quench should be evaluated in order to discuss the effect of SO coupling on the thermalization.

If the authors clearly solve the problem about the thermalization and they provide a conclusive evidence of the reduction of the thermalization, then I consider that the work includes enough important and interesting results for publication.

Other comments

(a) In Fig. 4(a), color bands (guides to the eyes) are not necessary. Difficult to see experimental data.

(b) The fit function to extract the decay time of the oscillations has the offset B_0 , but the offset B_0 should be zero. Why is the offset necessary? Related to this, there seem to be some offset for the GPE simulations of spin-dipole oscillations in Fig. S10. What is the reason for this?

(c) Information on the critical velocity of the superfluid might be helpful for the discussion on the thermalization.

Reviewer #1 (Remarks to the Author):

In the present paper the authors are presenting experimental results and theoretical analysis on the behaviour of the spin-dipole mode in a two-component Bose-Einstein condensate. In particular, after a quench in the Rabi-coupling of the spin-orbit lasers they observe the relaxation of the spin-dipole mode and the thermalization of the condensate fraction as a function of the final Rabi coupling after the quench. The experimental results are also compared with numerical simulations using the two-component Gross-Pitaevskii equation. The spin-dipole mode has been observed before, but what is new in this paper is the study of the effect of the spin-orbit coupling.

The paper is reasonable well written (but see below) and will be mostly interesting to the community working on ultracold gases. The authors claim that their results are of much broader interest, but that is not so obvious because in spintronics one is most often interested in fermionic (electronic) systems and here we are dealing with bosons. That community is presently also interested in magnon spintronics, which indeed deals with bosons, but the description is then based on the Landau-Gilbert equation, which is quite different from the Gross-Pitaevskii equation which is relevant for atomic Bose-Einstein condensates. It should be noted here that the Gross-Pitaevskii equation also does not include damping and thermalization, and so it is not expected that this description will be sufficient to understand all the details of this experiment. So presumably specific details of the atomic gas, such as the thermal cloud, will be important and this further reduces the overlap with the magnon systems.

Reply:

We agree with the reviewer that the Landau-Gilbert equation is used to describe magnons in many cases. On the other hand, we also notice that for magnon BECs, the Gross-Pitaevskii (GP) equation can be a good description in some cases. Please see e.g. [DOI: 10.1038/NPHYS3838](https://doi.org/10.1038/NPHYS3838) (now added as Ref [11] in the revised manuscript) as well as <https://arxiv.org/abs/1808.07407>, and [DOI: https://doi.org/10.1103/PhysRevB.81.020414](https://doi.org/10.1103/PhysRevB.81.020414). For exciton-polariton BECs in solid-state microcavities, the theoretical description is often based on the GP equation. Please see e.g. <https://doi.org/10.1103/RevModPhys.82.1489>. Therefore, we believe that our work may also be of interest to people studying bosonic quantum fluids in spintronics.

We agree with the reviewer that the GP equation, which does not include thermal components, is not sufficient to understand all the details of this experiment. However, in our case, the GP equation can still capture many important aspects of the physics including the SOC-enhanced damping of the spin-dipole mode (as we show in Fig. 6).

Having said that I believe this paper presents new and interesting results that deserve to be published. It is timely in the sense that it deals both with spin-orbit coupling and nonequilibrium transport in an atomic Bose gas. My main objection with this paper is its present form. I believe that the paper would be much more readable and would have much more impact if the authors would write a single text that

combines both the main text and the supplementary material. For instance, it is very important for this paper that the experimental results are compared with experiment, especially because the theory is expected not to be sufficiently accurate for all the experimental details. (For instance, there is no thermal cloud in the Gross-Pitaevskii equation, which will be crucial for damping and thermalization.) However, this comparison is only made in the supplementary material and not in the main text. Also the details of the fits of the shape of the clouds of the two spin components are crucial for understanding the final results on the thermalization and these are again only presented in the supplementary material. In my opinion, a combined text would be much more suited for publication and would be much helpful for the ultracold gases community, because the main text by itself does not give the necessary information to fully appreciate the results obtained by the authors.

Reply: We appreciate the reviewer's comment for significantly improving the presentation form of the paper. In light of the reviewer's suggestions, many figures with their corresponding texts presenting relevant information have now been moved from the supplementary materials to the main text:

- (1) New Fig. 5, showing the observation of deformation and shape oscillations of BECs.
- (2) New Fig. 6 is modified from the previous Fig. 5 to include the GPE-simulated SDM in addition to the real-space BEC density distributions at an early instant of the SDM. The extracted SDM damping compared with the experiment is also included.
- (3) New Fig. 7, showing the calculated nonorthogonality, effective interaction parameters, and immiscibility for the two dressed spin states.
- (4) New Fig. 8, showing the GPE-calculated different forms of energies per particle at different Raman couplings.

Reviewer #2 (Remarks to the Author):

Summary of the key results:

In the manuscript “Spin Current Generation and Relaxation in a Quenched Spin-Orbit Coupled Bose-Einstein Condensate”, the authors report on effects of synthetic SOC and spin transport using two interacting BECs of different spins states. These results are certainly worthy of publication in Nature Communications (with revisions). Mainly due to the substantive claim that SOC can significantly enhance SDM damping and reduce thermalization of BEC.

claims made:

- 1) Out-of-phase oscillations (spin-dipole mode, SDM) of the two BECs account for the spin current. — trivial claim
- 2) a) SOC can significantly enhance SDM damping and b) reduce thermalization of BEC — substantive claim
- 3) observe generation of diff BEC collective excitations, such as shape oscillations. — trivial claim
- 4) work may advance fundamental understanding of spin transport and quenched dynamics in SOCed systems.

The system consists of Rb-87 BECs in two hyperfine ground states that are then Raman coupled (Fig1b) to create synthetic 1D SOC with $|mF=-1\rangle$ and $|mF=0\rangle$ spin states. The BEC is initially prepared around the single minimum of the ground dressed band and after a hold time the final Raman coupling is quickly lowered to a final value that puts the BEC into the double minimum regime.

SDM oscillations are observed as a function of hold time at various final Raman couplings and show that at higher final Raman coupling, these oscillations are damped significantly which supports claim 2a. (Fig3 a-f). They further this claim and 2b in Fig4a-c by showing that less heating leads to an increase in condensate fraction.

To support claims 1), 2a-b), they complete control experiments:

Control exp1 - dipole oscillations of a SOCed BEC w single dressed spin comp: very small damping and negligible thermalization. — Spin current and net mass current

Control exp2 - exciting in-phase dipole oscillations of 2 dressed spin components of a SOCed BEC w/out relative collisions —> very small damping and negligible thermalization.— AC mass current with spin current.

In regards to claim 3) thermalization and spin current, It is not apparent in the text why the basic physics is happening. Adding a few lines to explain/describe such physics would benefit the non-expert

reader.

Reply: We appreciate the referee for considering “These results are certainly worthy of publication in Nature Communications (with revisions)”. We are not quite sure whether the reviewer asked about claim 2 (about thermalization and spin current) or claim 3 (about the generation of BEC shape oscillations) when he/she asked us to add a few lines to explain “why the basic physics is happening”. Therefore, we address each claim in the following:

For claim 2:

In our case, thermalization of the BEC is referred to as the reduction of the condensate fraction. Our experiments show that thermalization within the time of measurement requires the relative collision (i.e. SDM) between the two spin components. Therefore, if the SDM is more damped, meaning the relative motion and collision between the two spin components of the BEC stops earlier, the thermalization is also expected to be stopped earlier (small τ_{therm}), resulting in a higher saturation condensate fraction f_s . Indeed, we have observed that larger final Raman coupling Ω_F can stop the SDM earlier (please see Fig. 3) and give rise to a smaller τ_{therm} and a larger f_s (please see the measurements in Fig. 4b). Our explanation is addressed in the last few sentences in the first paragraph of the left column on page 5.

For claim 3:

As introduced in the main text, the total kinetic energy is the sum of three types of kinetic energy: quantum pressure (QP), local current kinetic energy (LC KE), and center-of-mass kinetic energy (CoM KE). QP and LC KE do not contribute to the BEC’s global translational motion which is accounted for by the CoM KE. Decreasing of the CoM KE in our case leads to the damping of the SDM. The SDM damping can be significantly enhanced due to the distortion of the BEC wavefunctions via converting the CoM KE into QP and LC KE (Fig. 8 in the revised manuscript). The increased QP and LC KE may lead to the emergence of excitations that do not have the CoM KE. This speculation is consistent with the observed generation of BEC shape oscillations, whose kinetic energy is accounted for by QP and LC KE rather than the CoM KE. In addition, the generation of BEC shape oscillations may also be understood by the observation of deformed atomic clouds (Fig. 5a-d in the revised manuscript) at early stages of the SDM, because the deformed shape is no longer in equilibrium with the trap and thus initiates the shape oscillations.

We have added a few lines in the left and right columns on page 7 in the revised manuscript and hoped that such physics becomes clearer.

claim 4) may contribute to vs advance... advance is a strong word here. Contribute to the understanding of quenched dynamics in SOCed systems, yes. Of course this is speculative, however, the final line of the conclusion, “may offer rich opportunities to study non-equilibrium quantum dynamics and novel superfluids in SOCed system”... is very vague.

Reply: Per referee's suggestion, we have changed the words to "Our work may contribute to the fundamental understanding of spin transport and quenched dynamics in spin-orbit coupled systems.". For the final line of the conclusion, it may have felt vague because we wanted the conclusion to be relatively general and brief. In light of referee's suggestions, we have added some examples and references to the sentence: "Experiments on SO coupled BECs, where many parameters can be well controlled in real time and with the potential of adding other types of synthetic gauge fields, may offer rich opportunities to study nonequilibrium quantum dynamics [67], such as Kibble-Zurek physics while quenching through quantum phase transitions [68], and novel superfluidity behaviors [16, 34] in SO coupled systems.".

Originality and significance: if not novel, please include reference:

Building off of prior methods, mainly refs [16,22], authors take it a step further and study spin current with two interacting BECs of different spins states. To my knowledge, this is the first study of in phase and out of phase oscillations of two interacting BECs demonstrating that Out-of-phase oscillations (spin-dipole mode, SDM) of the two BECs account for the spin current. This work has the added and substantive novelty that they understand a new mechanism for damping that is not of immediate intuitive transparency.

Data & methodology: validity of approach, quality of data, quality of presentation

The authors explain their methodology very clearly and openly discuss the caveats of their approach.

Appropriate use of statistics and treatment of uncertainties

Error bars on Raman coupling, condensate fraction and thermalization time seem reasonable, making the treatment of uncertainties clear.

Conclusions: robustness, validity, reliability

The data and supplemental documentation support the conclusions the authors have made.

Suggested improvements: experiments, data for possible revision

No additional experiments or data is needed to support the claims.

References: appropriate credit to previous work?

Yes. Perhaps the following paper is also a relevant reference: 2011a_ Generation of Dark-Bright Soliton

Trains in Superfluid-Superfluid.pdf

<https://arxiv.org/abs/1005.2610>

Reply: We thank the reviewer's suggestion and have cited this paper in Ref [39].

We have also cited a recent relevant paper (DOI: <https://doi.org/10.1103/PhysRevLett.120.170401>) about spin superfluidity in Ref [42].

Reviewer #3 (Remarks to the Author):

In this paper, effects of spin-orbit (SO) coupling on a spin current in a Bose-Einstein condensate (BEC) are investigated. By rapidly changing the strength of engineered SO coupling, the authors drive a spin current consisting of out-of-phase oscillations of two BECs of different spin states, and they experimentally observe the subsequent dynamics with and without SO coupling. From these measurements they conclude that the effects of SO coupling are (i) enhancement of damping of the spin-dipole oscillations and (ii) reduction of thermalization. The first effect (i) is confirmed by numerical simulations based on the time-dependent Gross-Pitaevskii equation. Furthermore, by investigating the time evolution of different contributions of energy as well as the spatial distribution of the BEC wavefunctions in their simulation, the authors reveal that the spatial modulation of the amplitude and phase of the wavefunctions is the reason for the strong damping of the spin-dipole oscillations.

The second effect (ii) sounds more interesting, because the strong damping is usually considered to contribute to the thermalization.

But here I am skeptical about the authors' conclusion. What they experimentally observed is that the BEC fraction becomes smaller for the dynamics without SO coupling. I simply considered that the change in SO coupling strength to trigger the dynamics is larger in the case of dynamics without SO coupling, leading to more energy input (and then more heating) to the system. The reduction of the BEC fraction somehow "normalized" by the energy input via quench should be evaluated in order to discuss the effect of SO coupling on the thermalization.

Reply: While it is true that quenching the Raman coupling to a lower final value Ω_F leads to more energy input to the system, it is important to point out that our experimental measurement of "thermalization" (reduction of condensate fraction) is performed only up to a relatively short time (~ 30 ms, after SDM damps out), *before* all of the initial energy input gets ultimately (presumably after much longer time) converted to the thermal energy (i.e., heating). In our relatively short time window, the quench leads to not only thermal excitations but also other excitations (e.g. collective excitations such as shape oscillations observed in our experiment), therefore initial energy input does not directly correspond to heating only, because it can also convert to other types of excitations.

It is also important to note that within the time of our measurements (the condition applied to all the following discussion), the thermalization in our case is caused by the collisions between the two spin components due to SDM, as shown in Fig. 4 and control experiments in Figs. S1 and S2 in the revised Supplementary Materials. These control experiments show that if there is no collision/SDM, then there is no observable thermalization even when there is still significant initial energy input. The additional observation from the red data (dipole oscillations of a single dressed spin component, excited from the same initial state to various Ω_F) in Fig. 3f further strengthens this conclusion. No notable

thermalization is observed even with the various energy input given, because there is no collision (i.e. no SDM). In addition, another control experiment has been performed for the SDM at $\Omega_F = 0$ using $t_E = 0.1 \text{ ms}$ to show that when there is SDM-induced thermalization, the SDM damping, not the energy input, is the key factor for thermalization. Compared to the case of $\Omega_F = 0$ with $t_E = 1.0 \text{ ms}$ in the main text, the case using $t_E = 0.1 \text{ ms}$ has the same energy input, but exhibits smaller SDM damping, longer thermalization time τ_{therm} , and smaller saturation condensate fraction f_s . Such a smaller SDM damping is attributed to a shorter t_E which gives shorter time of staying in finite Raman coupling before reaching $\Omega_F = 0$. Based on the above observations, we conclude that SDM (collision), not the energy input, is the key factor determining the thermalization behavior observed in our experiment.

In our case, thermalization is induced by the SDM. Therefore, the enhanced SDM damping observed at larger Ω_F causes the SDM to stop earlier and thus the thermalization (caused by the SDM) is also stopped earlier, leading to higher f_s . This expected thermalization behavior is indeed observed in our experiment (Fig. 4ab) and consistent with the above control experiments.

It is also worthwhile to point out that the quench mainly leads to a change of Raman energy (related to Ω_F , see Eq. (23) in the revised manuscript) which is the main contribution to the different initial energy input, see Fig. 8abef in the revised manuscript. Other types of energies, such as the kinetic energy, have very similar initial values at different Ω_F (where the change of momentum/vector potential due to $q_{\sigma min}$ are all similar) used in our experiment. Such Raman energy change also does not directly correspond to heating within our experimental time scale, again consistent with our explanations above.

If the authors clearly solve the problem about the thermalization and they provide a conclusive evidence of the reduction of the thermalization, then I consider that the work includes enough important and interesting results for publication.

Other comments

(a) In Fig. 4(a), color bands (guides to the eyes) are not necessary. Difficult to see experimental data.

Reply: We have removed the color bands to improve the data visibility.

(b) The fit function to extract the decay time of the oscillations has the offset B_0, but the offset B_0 should be zero. Why is the offset necessary?

Reply: The offset arises because we are only fitting the momentum of the major bare spin component (which decays to a *nonzero* momentum after the SDM damps) of each dressed spin state. More specifically, we determine $\hbar k_{spin} = \hbar(k_{\uparrow} - k_{\downarrow})$, where $\hbar k_{\uparrow} = \hbar(q_y - k_r)$ ($\hbar k_{\downarrow} = \hbar(q_y + k_r)$) is

the mechanical momentum of the **majority bare spin component** \uparrow (\downarrow) of the dressed spin state \uparrow' (\downarrow'). In the presence of SOC, after the SDM is damped, each dressed spin component (\uparrow' or \downarrow') relaxes to the corresponding band minimum at $q_y = q_{\sigma min} > 0$ (or $q_y = -q_{\sigma min}$), thus $\hbar k_{spin} = 2\hbar(q_{\sigma min} - k_r)$ at a finite Ω_F can be slightly nonzero because $q_{\sigma min} \neq k_r$. Thus, including the offset B_0 is needed in our fitting.

Related to this, there seem to be some offset for the GPE simulations of spin-dipole oscillations in Fig. S10. What is the reason for this?

Reply: Please see Fig. 6ab in the revised manuscript. In simulations, the mechanical momentum distribution of the atomic cloud is distorted and broadened. It is thus difficult to distinguish the majority and minority spin components of the same dressed spin state by their momentum difference. Thus, when we calculate the average mechanical momentum, say p_{\uparrow} of the atomic cloud associated with \uparrow , the calculated p_{\uparrow} would include a major contribution from the \uparrow component belonging to \uparrow' and a minor contribution from the \uparrow component belonging to \downarrow' . After the SDM is fully damped, such a calculation would lead to a momentum difference between the calculated p_{\uparrow} and p_{\downarrow} , manifested as some offset for the GPE-simulated SDM in Fig. 6c in the revised manuscript.

Nevertheless, the offsets mentioned above do not affect the conclusions of this paper.

(c) Information on the critical velocity of the superfluid might be helpful for the discussion on the thermalization.

Reply: The critical (relative) velocity for two counter-propagating SO coupled BECs (our case) is still an open question, and is different from the critical velocity of a SO coupled BEC against an *impurity*. It is important to point out that the traditional Landau criterion for the critical velocity is not applicable in SO coupled BECs due to break down of the Galilean invariance (see. e.g., our Ref. [34]). The most relevant experiment may be the measurement of the critical speed for the onset of the dynamical instability of a SO coupled BEC in a moving 1D lattice (DOI: <https://doi.org/10.1103/PhysRevLett.114.070401>). We note that the initial relative velocity between the two BECs (12 mm/s, Fig. 2) in SDM is notably larger than the estimated speed of zero sound (phonon-like excitations) $c_0 = \sqrt{\mu/m} \approx 2.3$ mm/s, which may be regarded as the upper scale of the critical velocity, where μ is the chemical potential and m is the atomic mass.

REVIEWERS' COMMENTS:

Reviewer #2 (Remarks to the Author):

The revised paper incorporating the previous main draft and supplemental material reads very well. The authors have addressed all of my questions and concerns at this time. I believe this paper presents new and interesting results that deserve publication.

Reviewer #3 (Remarks to the Author):

In the revised manuscript, the meaning of "thermalization" in this work is clearly explained. I agree with the authors' conclusion about the thermalization. Only my concern now is that the phrase of "reducing the thermalization of the BEC" in abstract is still misleading because there is no explanation of the meaning of "thermalization" here.

The authors addressed all the points raised by the referees, and the manuscript has been improved.

Response to the reviewers' comments:

Reviewer #2 (Remarks to the Author):

The revised paper incorporating the previous main draft and supplemental material reads very well.

The authors have addressed all of my questions and concerns at this time.

I believe this paper presents new and interesting results that deserve publication.

Reply: We appreciate the reviewer's comments and support for our work.

Reviewer #3 (Remarks to the Author):

In the revised manuscript, the meaning of "thermalization" in this work is clearly explained. I agree with the authors' conclusion about the thermalization. Only my concern now is that the phrase of "reducing the thermalization of the BEC" in abstract is still misleading because there is no explanation of the meaning of "thermalization" here.

The authors addressed all the points raised by the referees, and the manuscript has been improved.

Reply: We appreciate the reviewer's comments and support for our work. The meaning of thermalization in this work has now been clearly defined in the abstract of the revised manuscript.